Technical Report

# Metaboverse enables automated discovery and visualization of diverse metabolic regulatory patterns

Jordan A. Berg [1,9] ✉, Youjia Zhou[2,3,12], Yeyun Ouyang[1,9,12], Ahmad A. Cluntun [1], T. Cameron Waller [4], Megan E. Conway[5], Sara M. Nowinski[1,10], Tyler Van Ry[1,6,11], Ian George[1], James E. Cox [1,6,7], Bei Wang [2,3] & Jared Rutter [1,7,8] ✉

Metabolism is intertwined with various cellular processes, including controlling cell fate, influencing tumorigenesis, participating in stress responses and more. Metabolism is a complex, interdependent network, and local perturbations can have indirect effects that are pervasive across the metabolic network. Current analytical and technical limitations have long created a bottleneck in metabolic data interpretation. To address these shortcomings, we developed Metaboverse, a user-friendly tool to facilitate data exploration and hypothesis generation. Here we introduce algorithms that leverage the metabolic network to extract complex reaction patterns from data. To minimize the impact of missing measurements within the network, we introduce methods that enable pattern recognition across multiple reactions. Using Metaboverse, we identify a previously undescribed metabolite signature that correlated with survival outcomes in early stage lung adenocarcinoma patients. Using a yeast model, we identify metabolic responses suggesting an adaptive role of citrate homeostasis during mitochondrial dysfunction facilitated by the citrate transporter, Ctp1. We demonstrate that Metaboverse augments the user's ability to extract meaningful patterns from multi-omics datasets to develop actionable hypotheses.

Metabolism plays a central role in many biological processes, including cell fate decisions, protein homeostasis, stress responses, energy production, cell signalling, DNA replication and silencing, and more[1–13]. The rigorous study of metabolism, including the use of high-throughput transcriptomics, proteomics and metabolomics, has generated a systematic map of metabolic reactions and their constituents. Large consortium projects, such as the Kyoto Encyclopedia of Genes and Genomes[14,15], the Human Metabolome Database[16] and the Reactome Pathway Database[17–19], have helped formalize systematic maps of metabolism. These resources and tools provide a more holistic understanding of metabolism.

[1]Department of Biochemistry, University of Utah, Salt Lake City, UT, USA. [2]School of Computing, University of Utah, Salt Lake City, UT, USA. [3]Scientific Computing and Imaging Institute, University of Utah, Salt Lake City, UT, USA. [4]Division of Computational Biology, Department of Quantitative Health Sciences, Mayo Clinic, Rochester, MN, USA. [5]Department of Oncological Sciences, Huntsman Cancer Institute, University of Utah, Salt Lake City, UT, USA. [6]Metabolomics Core Facility, University of Utah, Salt Lake City, UT, USA. [7]Diabetes & Metabolism Research Center, University of Utah, Salt Lake City, UT, USA. [8]Howard Hughes Medical Institute, University of Utah, Salt Lake City, UT, USA. [9]Present address: Altos Labs, Redwood City, CA, USA. [10]Present address: Department of Metabolism and Nutritional Programming, Van Andel Institute, Grand Rapids, MI, USA. [11]Present address: College of Osteopathic Medicine, Michigan State University, East Lansing, MI, USA. [12]These authors contributed equally: Youjia Zhou and Yeyun Ouyang. ✉e-mail: jordanberg.contact@gmail.com; rutter@biochem.utah.edu

To circumvent the challenges associated with the inherent complexity of metabolic networks, it is common to adopt reductionist approaches. Although such strategies are vital to advancing our biological understanding, they can conceal multi-dimensional properties of metabolism as biological perturbations lead to complex, cooperative effects, many of which may seem negligible in isolation. Thus, reductionism can limit insight[20,21]. Additional challenges arise with the sparsity of metabolomics datasets[22,23] and the use of different network visualization parameters, which each influence how effectively one can interpret metabolic data[24].

To address these limitations, we created Metaboverse, an interactive application for exploring multi-omics data in the context of the metabolic network and for generating data-driven hypotheses. Metaboverse delivers four critical innovations or contributions that provide a powerful interface for the interpretation of data. First, Metaboverse uses a diverse library of possible metabolic patterns to search the metabolic network. Second, Metaboverse generates summarized reaction representations that span multiple reactions, enabling the discovery of patterns across sparse datasets and between pathways. Third, Metaboverse automates pre-processing and network curation tasks for a diverse set of model organisms. Fourth, Metaboverse enhances the contextualization of these patterns by providing a dynamic exploratory interface to facilitate hypothesis generation.

In this Technical Report, we detail these components and their integration into Metaboverse, present two vignettes in which we use Metaboverse to analyse public and newly generated datasets, and outline important patterns that were detected using Metaboverse but not by existing methods. We will present benchmarks between Metaboverse and other comparable tools, as well as sensitivity analysis detailing Metaboverse pattern recognition with sparse datasets. Metaboverse is available at https://github.com/Metaboverse.

## Results

### Dynamic reaction visualization augments hypothesis creation

Metaboverse curates a reaction network database on the basis of a Reactome knowledgebase[17–19], BiGG[25] or BioModels[26,27] network. Users provide any combination of transcriptomics, proteomics and/or metabolomics data, which are then integrated into the reaction network as $\log_2$(fold change) and statistical values for each measurement and sample comparison (Extended Data Fig. 1). An interactive data formatting aid is available for users requiring assistance to format their data for Metaboverse. Additional methods during data processing allow for the interpolation of protein complex measurements or protein measurements from upstream components, such as protein or gene measurements.

Once Metaboverse integrates user data onto the network, interactive tools help visualize and explore reactions and their components individually, by canonical pathway definitions or by nearest reaction neighbourhood networks for any given network component. Nearest reaction neighbourhood networks consequently aid in identifying upstream or downstream patterns that may occur between pathways[24]. We integrated visualization options to limit the display of metabolic hubs as detailed in our previous work[24] to assist the user's exploration of the data beyond the most familiar pathways.

### Pattern recognition enables robust data interpretation

Previously, we described MetaboNet and DyMetaboNet, wherein we identified multi-dimensional, reaction-based patterns by manually comparing inputs and outputs of reactions and looking for general trends across a reaction[24]. While this approach identified several interesting patterns within datasets, it was time consuming and incomplete. Similarly, other existing tools may provide some pattern identification capabilities, but the scope of the patterns they can capture is limited and fails to capture vital metabolic signatures (Supplementary Note 1 and Supplementary Table 1).

Metaboverse introduces a broad collection of algorithms that enable the rapid and automated discovery of various complex regulatory patterns (Extended Data Fig. 2a), including using information about reaction catalysts and inhibitors to identify more complex patterns. Generally, Metaboverse compares the inputs (substrates), outputs (products) and modifiers (catalysts and inhibitors) of each reaction to determine if there is a net change across the reaction (Supplementary Notes 2 and 3, equations (1)–(11)). It then returns an interactive list of reactions that pass the specified fold change and/or statistical thresholds. Users may also select annotated pathways in which that reaction is found for exploration and contextualization.

### Data sparsity and the identification of complex patterns

Missing data points, particularly in metabolomics experiments, are frequent and make the identification of network regulatory patterns challenging[22,23]. Over 1,000 metabolites are known to participate in human metabolism, yet current metabolomics technologies limit the number of metabolites that are typically quantified. This results in gaps in the measured metabolic network and can confound pattern recognition across reactions. We developed algorithms that collapse up to three connected reactions with intermediate missing data points if they can be bridged with measured data on the distal ends of the reaction series (Extended Data Figs. 2b and 3 and Methods). Thus, if an intermediate reaction component is unmeasured in the user's dataset, but an input to the reaction producing the unmeasured component or an output to the reaction consuming the unmeasured component was quantified, these two reactions can be combined into one representative reaction. Similar concepts have been used to define amino acid-related metabolites[28].

### Metabolic signatures in LUAD

We first used public steady-state metabolomics data from early stage human lung adenocarcinomas (LUAD)[29] to test whether Metaboverse could capture the metabolic perturbations identified in this study as well as additional signatures. In this specific study, the authors asked which metabolites could be used as diagnostic markers to identify early stage adenocarcinomas (Fig. 1a). Metaboverse reliably prioritized patterns in nucleotide metabolism using collapsed and uncollapsed reaction representations (Fig. 1d–g and Extended Data Fig. 4) that were consistent with the original study[29] and our manual re-analysis of the data using MetaboNet and DyMetaboNet[24]. Metaboverse also identified reaction patterns related to xanthine metabolism, a metabolite highlighted in the original study[29] (Fig. 1d,g and Extended Data Fig. 4a), and a collapsed reaction pattern driven by a reduced relative abundance of citric acid and an increased relative abundance of both glutamic acid and malic acid, with cross-pathway connections to a decrease in lysine (Fig. 1b,c and Extended Data Fig. 4b). Although changes in the concentrations of these metabolites were identified in the original study, their relevance to the study cohort and their connectedness within the metabolic network were not discussed[29].

Comparing the substrate and product measurements of a reaction allows us to identify multi-dimensional patterns that provide further insight into metabolic behaviours[24]. Strikingly, the top-ranking `Average` reaction pattern (Supplementary Note 3, equation (2)) suggested spermine synthase activity (SMS) (Fig. 2a), catalysing a reaction connected to polyamine synthesis. The second top-ranking `Average` reaction pattern (Supplementary Note 3, equation (2)) suggested the activity of glycerate kinase (GLYCTK) (Extended Data Fig. 5), a pattern identified in our previous manual re-analysis of the data[24], which could implicate perturbations in serine metabolism—a pathway that contributes to tumorigenesis[30]. These particular connections were missed in the original study but were quickly and automatically highlighted by Metaboverse.

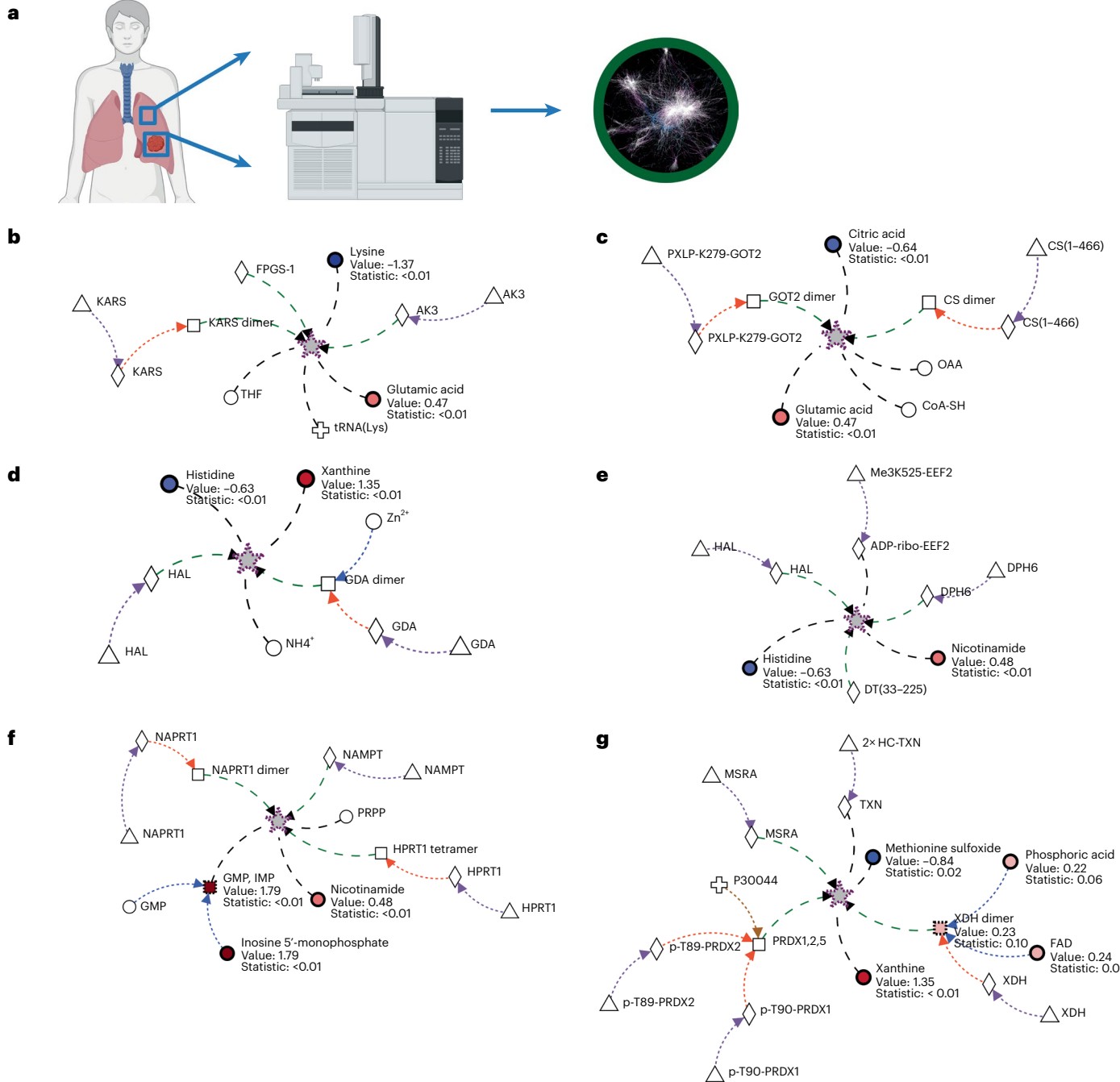

**Fig. 1 | Metaboverse identifies key regulatory signatures in early stage LUAD steady-state metabolomics data using collapsed reaction representations. a**, Concept map of the study analysed by Metaboverse. Paired normal and tumour lung samples were acquired from each patient and MS was performed to quantify metabolite abundances (Metabolomics Workbench project PR000305; subpanel created with BioRender.com). **b–g**, Representations of six top-ranking, non-redundant collapsed reaction patterns. Dashed-black edges indicate connections between distal ends of two to three reactions that were collapsed. Stars with a dashed-purple border indicate a collapsed reaction. Collapsed reactions were identified using the `Average` reaction pattern. **b**, Lysine + tRNA(Lys) + ATP → Lys-tRNA(Lys) + AMP + pyrophosphate // Adenylate Kinase 3 is a GTP-AMP phosphotransferase // mitochondrial FPGS-1

transforms THF to THFPG. **c**, Acetyl-CoA + $H_2O$ + oxaloacetate → citrate + CoA // oxaloacetate + glutamate ↔ aspartate + alpha-ketoglutarate (GOT2) (cross-pathway). **d**, Histidine → urocanate + $NH_4^+$ // guanine + $H_2O$ → xanthine + $NH_4^+$ (cross-pathway) **e**, Histidine → urocanate + $NH_4^+$ // DPH6 ligates ammonium to diphthine-EEF2 // DT fragment A ADP-ribosylates target cell EEF. **f**, HPRT1 catalyses the conversion of guanine or hypoxanthine to GMP or IMP // NAPRT1 dimer transfers PRIB to NCA to form NAMN // NAMPT transfers PRIB to NAM to form NAMN (cross-pathway) **g**, XDH oxidizes xanthine to form urate // PRDX1,2,5 catalyse TXN reduced + $H_2O_2$ → TXN oxidized + $2H_2O$ // MSRA reduces L-methyl-(S)-S-oxide to L-methionine. *P* values were derived using a two-tailed, homoscedastic Student's *t*-test and adjusted using the Benjamini−Hochberg correction procedure. Source numerical data are available at ref. 55.

## Metaboverse signatures correlate with patient prognosis

To assess whether the reaction patterns identified by Metaboverse could provide meaningful insights into the clinical outcomes, we analysed a LUAD cohort from The Cancer Genome Atlas (TCGA) using

Cox regression analysis with an optimized expression cut-off[31–33], and observed striking correlations between the expression of the gene encoding SMS and patient survival outcomes (optimized FPKM (fragments per kilobase of transcript per million) cut-off: 49.5413; Fig. 2b

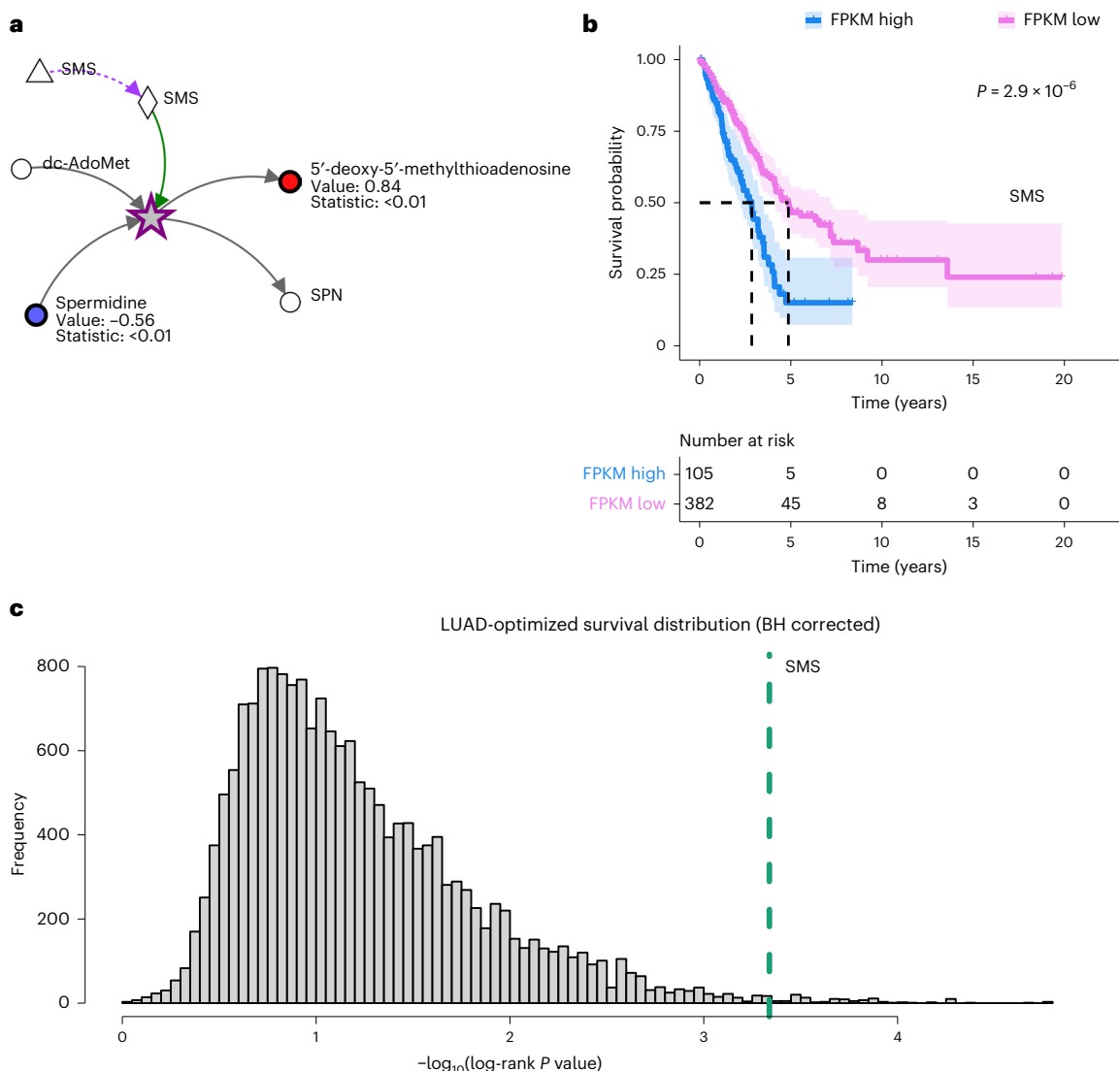

**Fig. 2 | Metaboverse identifies and contextualizes a putative regulatory signature in early stage LUAD steady-state metabolomics data. a,** Reaction between spermidine and 5‑methylthioadenosine identified by Metaboverse's reaction pattern recognition module as the highest-ranking `Average` reaction pattern. Metabolomics values are shown as node shading, where an increasingly blue shade indicates decreased abundance compared with normal tissue, and an increasingly red shade indicates increased abundance compared with normal tissue. Measured $\log_2$(fold change) and Benjamini–Hochberg corrected $P$ values for each entity are displayed below the node name. Green edges indicate a catalyst. **b,** Kaplan–Meier plot of Cox regression analysis for the optimal expression cut-off calculated for *SMS* (FPKM cut-off: 49.5413; high: 105 tumours; low: 382 tumours). Shading indicates 95% confidence intervals for each expression group. Dashed lines indicate the median survival times for each group. Risk tables are displayed below the plot. $P$ value was derived from the log-rank test. **c,** Distribution of Benjamini–Hochberg-corrected log-rank $P$ values for the Cox regression of each gene in the TCGA–LUAD RNA-seq cohort. *SMS* is indicated by the dashed-green line. Source numerical data are available at ref. 55.

and Extended Data Fig. 5). Notably, the log-rank $P$ value for *SMS* gene expression ranked in the top 0.65% of all regressions (118 of 18,169 surveyed genes) (Fig. 2c).

Alternatively, the log-rank $P$ value for *GLYCTK* gene expression in the LUAD cohort was poor (optimized FPKM cut-off: 0.913; Extended Data Fig. 5), possibly due to the lower DepMap dependency score of *GLYCTK* compared with *SMS* gene expression in LUAD cell lines (Extended Data Fig. 6a) (refs. 34,35). Additionally, at the time of writing, SMS is the only spermine-producing enzyme annotated in humans (Reactome), whereas the product of the reaction catalysed by the GLYCTK enzyme, 3-phosphoglyceric acid, can also be produced by the glycolytic enzyme phosphoglycerate kinase 1 (PGK1). It is also interesting to note the modest correlation between the top-ranking LUAD reaction pattern enzymes and their corresponding survival statistics in LUAD gene expression data (Extended Data Fig. 6b).

SMS has been implicated in the silencing of *Bim*, which encodes a pro-apoptotic factor, in colon adenocarcinomas[36], and *SMS* gene expression tends to correlate with more proliferative cell types in the lung (Extended Data Fig. 6c) (refs. 36,37). Similar patterns in SMS have also been identified in mouse xenografts of lung cancer[38]. Thus, Metaboverse-guided predictions may inform research directions and treatment strategies for this disease.

## Metabolic signatures of respiratory impairment in yeast
To further demonstrate Metaboverse in the context of multi-omics and time course or multi-condition datasets, we analysed a model of mitochondrial fatty acid synthesis (mtFAS) deficiency in *Saccharomyces cerevisiae*. mtFAS is an evolutionarily conserved pathway that produces lipoic acid, a critical co-factor for many metabolic enzymes. Recent work has uncovered additional biological roles for this pathway,

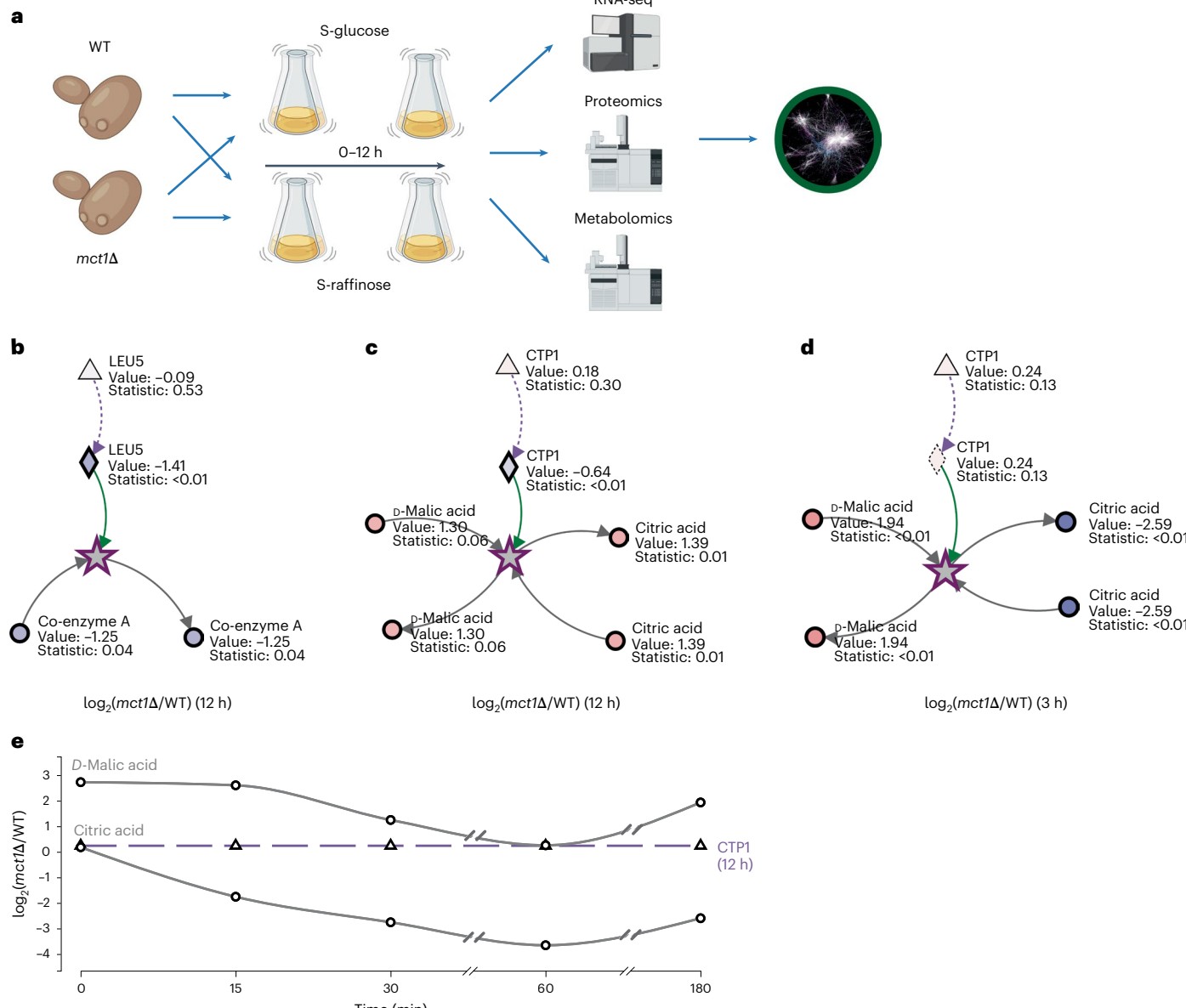

**Fig. 3 | Metaboverse identifies signatures of mitochondrial dysfunction from multi-omics data in yeast. a**, Concept map of the study analysed by Metaboverse. Yeast were pre-incubated in S-glucose media and then grown over a time course in either S-glucose or S-raffinose media. Samples were analysed using RNA-seq (*n* = 4), proteomics (*n* = 3) and metabolomics (*n* = 6, except for the 3 h WT group, where *n* = 5) (subpanel created with BioRender.com). **b–d**, Reaction pattern catalysed by Leu5 (**b**) and Ctp1 (**c** and **d**) using late (12 h) RNA-seq, proteomics and metabolomics data (**b** and **c**) or early (3 h) RNA-seq and

metabolomics data (**d**). Reaction pattern graph values are shown as node shading, where an increasingly blue shade indicates decreased abundance compared with WT, and an increasingly red shade indicates increased abundance compared with WT. Measured log$_2$ (fold change) and statistical values for each entity are displayed below the node name. **e**, Log$_2$ fold changes of citrate, malate and Ctp1 across timepoints. Benjamini–Hochberg corrected *P* values were used for proteomics and metabolomics data and false discovery rate was used for RNA-seq data. Source numerical data are available at ref. 55.

including in the generation of acylated-mitochondrial acyl carrier protein, which controls the assembly and activation of mitochondrial oxidative phosphorylation complexes[39–41]. The discovery of patients with mutations in genes encoding key mtFAS enzymes further illustrates the physiological importance of this pathway[42]. Deletion of the *MCT1* gene (homologous to the Malonyl-CoA-Acyl Carrier Protein Transacylase in humans) abolishes the activity of the mtFAS pathway[40], allowing us to use an *mct1Δ* mutant to determine the effects of mtFAS pathway perturbation on metabolism.

We utilized previously generated proteomics data in *mct1Δ* yeast after a shift from glucose- to raffinose-supplemented growth media, which triggers the biogenesis of mitochondria and tends to pronounce

respiratory defects[40], together with RNA sequencing (RNA-seq) at 0, 3 and 12 h and steady-state metabolomics at 0, 0.25, 0.5, 1, 3 and 12 h after this shift in growth media (Fig. 3a). By layering these multi-omics data onto the *S. cerevisiae* metabolic network, we could explore acute and chronic responses to mtFAS deficiency.

The top-ranked `ModReg` and `TransReg` reaction patterns (Supplementary Note 3, equations (4) and (5)) at 12 h predictably centred around the abundance of mitochondrial carrier proteins, Coenzyme A-related reactions and iron–sulfur cluster biogenesis (Fig. 3b and Extended Data Fig. 7a–c,e). We also observed respiratory signatures consistent with previous studies[40,43], including the identification of patterns in electron transfer from ubiquinol to cytochrome C via Complex

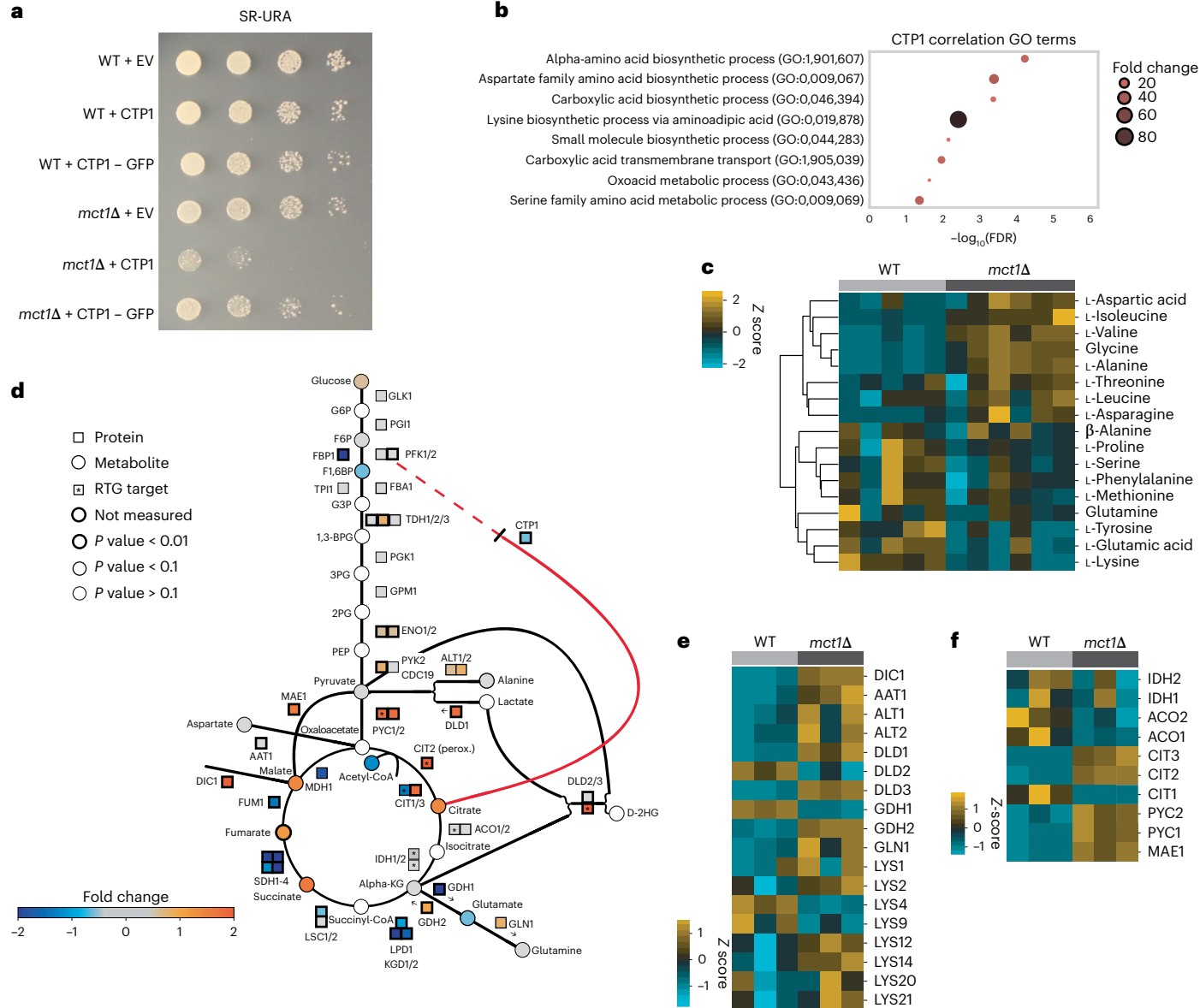

**Fig. 4 | Metaboverse-identified signatures suggest compensatory mechanisms to mitochondrial dysfunction in yeast. a**, Spot growth assays for WT and *mct1Δ* mutant strains with EV, overexpression of *CTP1* and overexpression of *CTP1*–GFP construct on SR-URA. **b**, GO term enrichment results for genes identified in the SpQN-corrected co-expression analysis of *CTP1* in the refine.bio WT cohort (*n* = 1,248). **c**, Heat map of amino acids for WT and *mct1Δ* mutant strain proteomics at 3 h after the switch to raffinose from glucose. **d**, Graphical overview of yeast glycolysis pathway and other related reactions overlaid with summary annotations based on RNA-seq, proteomics and

metabolomics measurements during steady-state growth (12 h). RTG, retrograde signaling pathway. A more complete legend for the shading criteria can be found in Extended Data Fig. 8. **e,f**, Heat maps of amino acid-regulated enzymes (**e**) and anaplerotic enzymes (**f**) for WT and *mct1Δ* mutant strain proteomics at 12 h after the switch to raffinose from glucose. Heat map values were mean centred at 0 (*z* score). Hierarchical clustering was performed where indicated by the linkage lines using a simple agglomerative (bottom-up) hierarchical clustering method (or UPGMA). Source plate images and numerical data are available at ref. 55.

III of the electron transport chain (ETC) (Extended Data Fig. 7g). We have observed similar patterns using mtFAS perturbation models in the past[39,40].

The second expected pattern of interest was the general reduction in the abundances of tricarboxylic acid (TCA) cycle-related enzymes and intermediate metabolites (Extended Data Fig. 7d). We observed an increase in Dic1 protein abundance (Fig. 4e and Extended Data Fig. 8), consistent with reports that *DIC1* gene expression may be essential for growth in the presence of certain carbon sources due to its role in shuttling phosphate across the mitochondrial inner membrane in exchange for malate or succinate[44]. Yeast with respiratory defects may adapt by increasing Dic1 protein levels to facilitate substrate transport

to maintain TCA cycle flux and mitochondrial respiration. Indeed, we found that whole-cell malate levels were elevated in *mct1Δ* mutants relative to wild type (WT) (Extended Data Figs. 8 and 9l).

**Metaboverse signatures correlate with metabolic adaptations**
One unexpected top-ranking reaction pattern identified using the 12 h multi-omics datasets involved the tricarboxylate transporter, Ctp1, which transfers citrate across the mitochondrial inner membrane (Fig. 3c and Extended Data Fig. 7h) (ref. 45). This reaction was also identified at the 3 h timepoint using metabolomics and RNA-seq data (Fig. 3d). Earlier metabolomics measurements showed citrate levels initially decreasing then increasing over the time course (Fig. 3e).

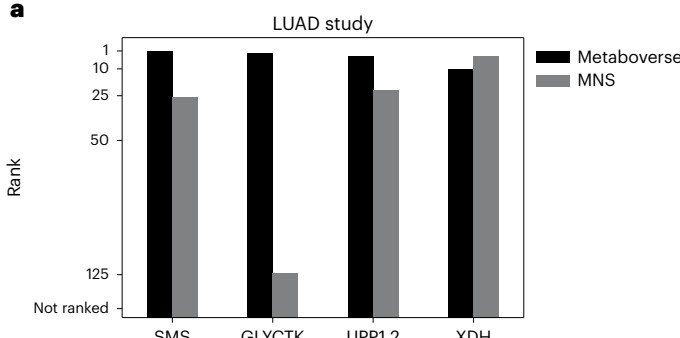

**a**

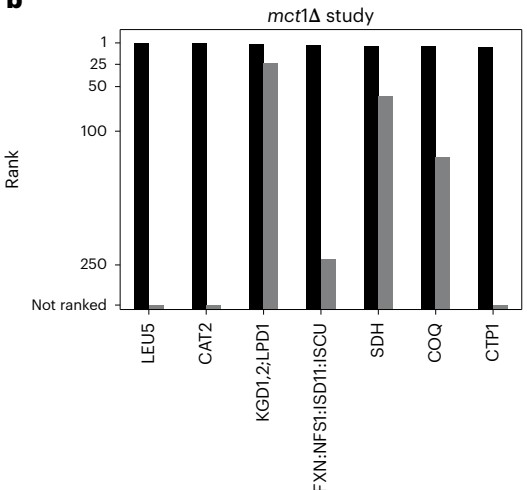

**b**

**Fig. 5 | Metaboverse prioritizes relevant, testable metabolic reaction patterns. a**, Top reaction patterns identified by Metaboverse that are supported by additional data from TCGA project compared with MNS rankings. **b**, Top reaction patterns identified by Metaboverse that are supported by previous validation or validation as a part of this manuscript compared with MNS rankings. X-axes represent the relevant genes' rankings in either Metaboverse or MNS, and y-axes represent the rank of each of the represented genes. Source numerical data are available at ref. 55.

Since citrate is a key metabolite in the TCA cycle, we hypothesized that Ctp1 protein levels decrease in response to early respiratory stresses in mtFAS-deficient cells.

We hypothesized that if *mct1*Δ cells adaptively downregulate Ctp1, then overexpression of Ctp1 should cause growth defects. Indeed, we observed a specific sensitivity to Ctp1 overexpression in the *mct1*Δ background (Fig. 4a and Extended Data Fig. 10a). We found that over-expression of a C-terminal green fluorescent protein (GFP)−*CTP1* fusion vector ablated this growth defect, and verified that this did not result from spurious effects from protein overexpression, such as the forma-tion of inclusion bodies (Fig. 4a and Extended Data Fig. 10b). These data suggest that the GFP−Ctp1 is inactive, mislocalized or otherwise perturbed, and the observed growth defect was due to functional Ctp1 localizing to the mitochondria[46–48].

We performed co-expression analysis of *CTP1* gene expression using thousands of uniformly processed WT yeast RNA-seq experi-ments[49,50] and discovered that correlating gene sets (SpQN-normalized Pearson's *r* > 0.5) included programmes related to the biosynthesis of aspartate, lysine and other amino acids (Fig. 4b). Aspartate can be con-verted into fumarate, which might partially explain increased fumarate concentrations despite an impaired ETC, while lysine can be used as a substrate for the generation of acetyl-CoA, which plays a central role in the mtFAS pathway[40]. Metabolic rewiring of these and other amino

acids was also apparent in the metabolomics (Fig. 4c) and proteomics datasets (Fig. 4e), potentially explaining the ability of Mct1Δ mutants to maintain growth despite metabolic dysfunction (Fig. 4a). However, *CTP1* gene overexpression appears to disrupt this biosynthetic rewir-ing, disabling the ability of *mct1*Δ cells to tune their growth (Fig. 4a).

We observed increased protein abundance levels in components of anaplerotic pathways, namely Mae1, Pyc1, Pyc2, Cit2/Cit3 and Gdh2 (Fig. 4f). These enzymes sequentially catalyse the conversion of malate to pyruvate to oxaloacetate to citrate, respectively. *PYC1* and *CIT2* are also targets of the retrograde signalling pathway, which elicits mitochondrial-to-nuclear communication during mitochondrial stress[51]. Work on the retrograde pathway has suggested that elevated Cit2 expres-sion functions to maintain metabolite pools for anabolic growth[51], which we also see in the *mct1*Δ (Fig. 4f and Extended Data Fig. 10). Pyruvate dehydrogenase levels are reduced in mtFAS-deficient cells, so upregu-lation of Pyc1 and Pyc2 probably provides an alternative pathway for converting pyruvate to oxaloacetate, which can then be converted into citrate for biosynthetic fuel[52,53] (Source Data Fig. 4).

## Metaboverse identifies meaningful and verifiable patterns

We evaluated how Metaboverse can offer a more comprehensive expe-rience when analysing metabolism-related datasets for known and novel regulatory patterns compared with existing metabolic network exploration tools. The scope and performance of many of these tools are summarized in Supplementary Note 1 and Supplementary Table 1. However, we seek to emphasize that the key findings and hypotheses we present in this manuscript regarding *SMS* in LUAD and *CTP1* in yeast were uniquely discovered, contextualized and/or prioritized using Metaboverse. For benchmarking purposes, we decided to focus our analysis on Metabolic Network Segmentation (MNS[54]) and Ingenuity Pathway Analysis (IPA; Qiagen), the two options with operable code or a graphical user interface at the time of writing (Supplementary Table 1).

We evaluated the ability of MNS to prioritize verifiable or canonical signatures within the LUAD metabolomics dataset (Figs. 1 and 2). MNS was able to identify both *SMS* (ranked number 27 in MNS and number 1 in Metaboverse) and *GLYCTK* (ranked number 125 in MNS and number 2 in Metaboverse), but their relevance to the dataset and their roles within the metabolic network were opaque. While MNS identified xan-thine dehydrogenase (*XDH*, ranked number 4 in MNS), Metaboverse, ranked this reaction as number 11 due to a poorer statistical value for hypoxanthine (Benjamini−Hochberg corrected *P* = 0.11) (Fig. 5a). Another challenge we experienced in using this software was the sheer number of genes per result, ranging from 1 to 447 for the top ten results from the dataset. The scope of these lists can greatly hamper the user's ability to generate actionable hypotheses (Source Data).

Next, we evaluated MNS prioritization of verifiable or canoni-cal signatures from the *mct1*Δ model (Figs. 3 and 4). We were only able to integrate single timepoint metabolomics data from the *mct1*Δ experiments as MNS does not have any multi-omics analysis capabili-ties (we chose to evaluate the 12 h timepoint). ETC defects are strong hallmarks of mtFAS dysfunction[40] and were successfully prioritized by Metaboverse but not by MNS (see KGD1,2; LPD1; FXN:NFS1:ISD11:ISCU; SDH in Fig. 5b, which range from hit 24 to 243 in MNS). While other citrate-related reactions were prioritized by MNS ('CIT2; CIT1; CIT3', 1 in MNS; 'ACO1', 2 in MNS; 'PFK1; PFK2; FBP1', 12 in MNS), their relation-ship to the transport reaction catalysed by *CTP1*, which we verified as a contributor to biogenesis during mitochondrial dysfunction, was missed (Figs. 3 and 4).

When comparing Metaboverse to IPA, we used the LUAD meta-bolomics data described previously[29]. The IPA graphical summary included 'Concentration of phosphotidylcholine', 'PPARGC1B' and 'ARNT', with no further context. IPA's pattern identification tools, 'Upstream regulators' and 'Causal networks', identified most of the *SMS*- and *GLYCTK*-related metabolites. However, IPA did not identify either enzyme (*SMS* or *GLYCTK*) and the 'Upstream

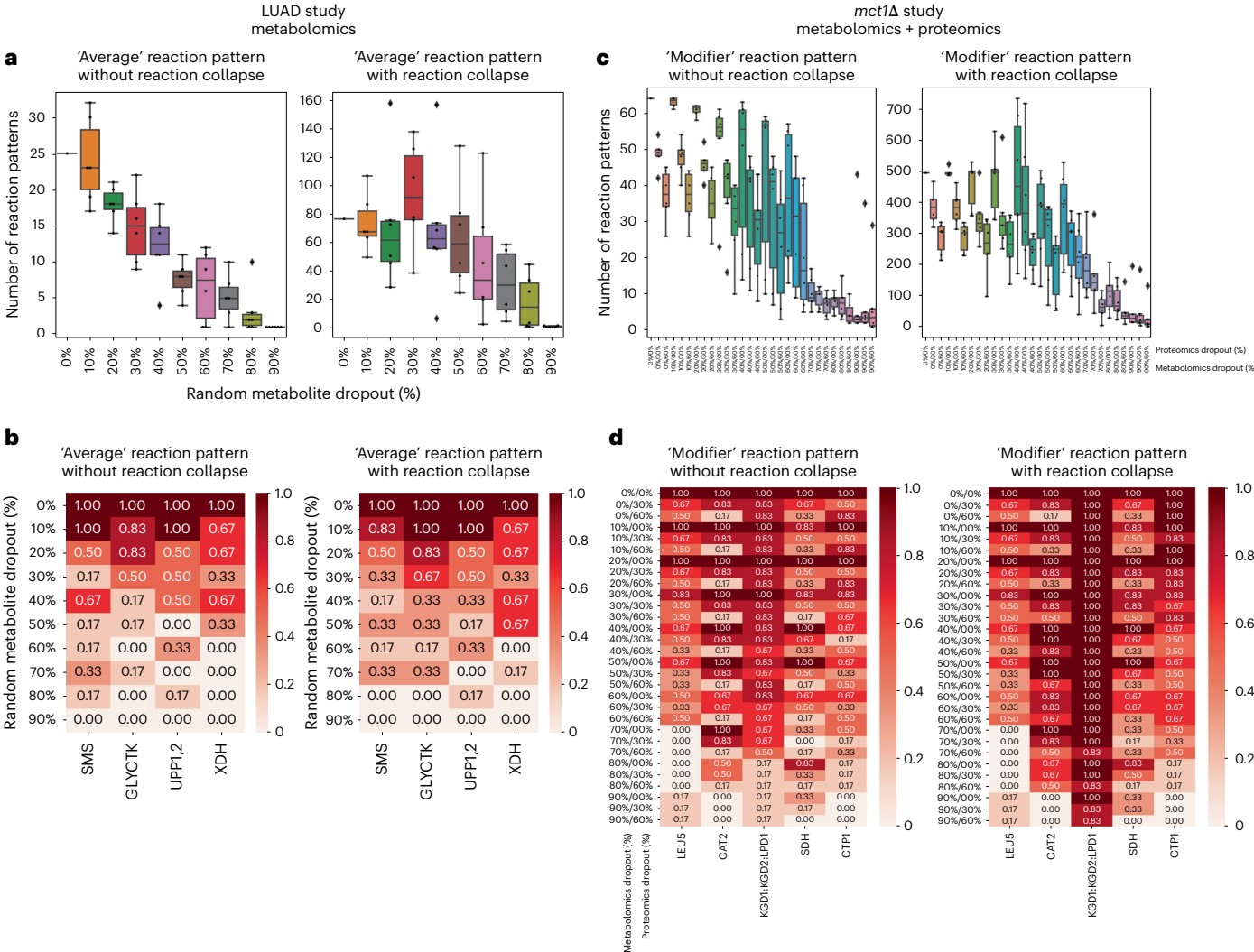

**Fig. 6 | Metaboverse pattern recognition is resilient to missing data.**
**a**–**d**, Random analyte dropout datasets for each omics type (*n* = 6 per dropout).
**a**, Box plots of the number of `Average` reaction patterns Metaboverse could identify within the LUAD metabolomics dataset with or without reaction collapsing with 0–90% of the original input metabolomics data missing.
**b**, Heat maps of the proportion of replicates that identified each of the signature reaction patterns for the LUAD dataset, as described as part of Fig. 5a. **c**, Box plots of the number of `Modifier` reaction patterns Metaboverse could identify

within the *mct1*Δ dataset within this study with or without reaction collapsing with 0–90% of the original input metabolomics data missing and 0%, 30% or 60% of the original input proteomics data missing. **d**, Heat maps of the proportion of replicates that identified each of the signature reaction patterns for the yeast dataset, as described as part of Fig. 5b. Box plot centre lines indicate mean, box limits indicate upper and lower quartiles, whiskers indicate 1.5× interquartile range, diamonds indicate outliers and dots indicate replicates. Source numerical data are available at ref. 55.

regulators' analysis identified spermidine 184 times, spermine 0 times, 5′-methylthioadenosine 31 times, glyceric acid 1 time (but as regulator number 613) and 3-phosphoglyceric acid 65 times. 'Causal networks' analysis likewise identified spermidine 160 times, spermine 0 times, 5′-methylthioadenosine 31 times, glyceric acid 0 times and 3-phosphoglyceric acid 56 times. In either case, these instances rank from 1 to 934 out of 940 total results, and within large aggregate lists of other metabolites, which resemble overly general set enrichment analysis. This approach, therefore, requires non-trivial parsing of large lists to identify relevant patterns.

**Metaboverse pattern analysis is robust against sparse data**
To evaluate the ability of Metaboverse to identify relevant metabolic signatures under conditions of data sparsity, a frequent hallmark of metabolomics data[22], we randomly removed 0–60% of quantified proteins and 0–90% of quantified metabolites from the two data vignettes presented herein. For each dataset, we highlighted the reaction pattern

type where the most relevant patterns were found in the Metaboverse analyses (the `Average` pattern for the LUAD study and the `ModReg` and `TransReg` patterns for the *mct1*Δ study).

The LUAD metabolomics study generated 183 consistent metabolite quantifications between paired adenocarcinoma and adjacent normal tissue[29]. We first evaluated the total number of possible reaction patterns (Methods) and found that the LUAD dataset contained 2,160/3,853 (56.06%) and 239/2,219 (10.77%) metabolic reactions with enough measured data with or without reaction collapsing, respectively. By randomly dropping out varying numbers of the 183 metabolites, we noticed a consistent decline in the number of reaction patterns Metaboverse was able to identify without collapsed reaction representations (Fig. 6a). With collapsed reaction representations, Metaboverse was more resilient against data sparsity (Fig. 6b). Metaboverse sometimes identified more reaction patterns with missing data (Fig. 6a). In these cases, reactions with more than one measured input or output can be detrimentally weighted by low values, thus lowering the reaction's

score. This emphasizes the importance of exploring the data with a variety of the different reaction pattern types provided by Metaboverse to capture the breadth of metabolic signatures.

Applying this evaluation design to the *mct1Δ* 12 h proteomics and metabolomics datasets, which contained 662/986 (67.14%) and 172/604 (28.48%) metabolic reactions with enough measured data with and without reaction collapsing, respectively. We again observed a steady decline in the number of identified reaction patterns. However, similar to the LUAD study, the use of collapsed reaction representations provided more resilience in the number of identified patterns (Fig. 6c). Strikingly, we observed that proteomics data appears to buffer Metaboverse pattern recognition despite missing data points (Fig. 6d). Reassuringly, in some cases, reaction patterns could be identified 100% of the time with up to 60% of the original metabolomics or proteomics data missing (Fig. 6d).

## Discussion

Metaboverse provides an easy-to-use interactive visualization tool for exploring multi-omics data in the context of the metabolic network and guiding hypothesis generation (Extended Data Fig. 1). Importantly, Metaboverse introduces pattern recognition and data sparsity handling algorithms to identify metabolic signatures within a user's dataset (Extended Data Fig. 2).

Using public metabolomics data from LUADs, we demonstrated that Metaboverse could not only identify reaction patterns corresponding to known metabolic signatures (Fig. 1), but it identified biologically relevant patterns that were undetected by existing data-driven approaches. For example, Metaboverse identified a previously undescribed reaction pattern centred around SMS, and we subsequently determined a positive correlation of LUAD patient survival with *SMS* gene expression (Fig. 2).

Using multi-omics datasets in a yeast model of mtFAS dysfunction, we demonstrated that Metaboverse could identify important regulatory and compensatory mechanisms, including adaptations to respiratory defects that enable these mutants to maintain their growth (Figs. 3 and 4). Specifically, the insights from Metaboverse implicate altered Ctp1 activity and citrate homeostasis as part of the broader biosynthetic response to mitochondrial dysfunction, allowing for continued cellular growth and survival.

We benchmarked Metaboverse against existing tools with similar goals to Metaboverse and found that Metaboverse consistently ranked relevant reactions higher and provided an intuitive format to explore these patterns (Fig. 5). We tested the ability of Metaboverse to identify critical reaction patterns with missing data points and found that the number of identified reactions decreased with increasing data sparsity, but this was buffered by the use of collapsed reaction representations and multi-omics data integration (Fig. 6).

Metaboverse integrates multiple data layers onto the metabolic network across model organisms, allowing users to easily analyse data to identify interesting patterns. Thus, Metaboverse provides an integrated platform for pattern recognition and hypothesis generation, and can assist users in their design of future experiments with a more holistic mindset.

## Online content

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

## Methods

### Network curation

Biological networks are curated using the current version of the Reactome knowledgebase[17–19]. In particular, the pathway records and Ensembl- and UniProt-Reactome mapping tables are integrated into the network database for Metaboverse. Additionally, the ChEBI and The Human Metabolome databases are also referenced for metabolite synonym mapping to accept more flexible metabolite input nomenclature from the user[16,56]. These data are used to generate a series of mapping dictionaries for entities to reactions and reactions to pathways for the curation of the total reaction network. Reaction annotations are additionally obtained from the Reactome knowledgebase[17–19]. At the time of writing, users can also provide BiGG[25] and BioModels[26,27] networks; however, full support cannot always be guaranteed due to the more bespoke nature of some network models from these sources. The resulting curation file is output as a pickle-formatted `.mvdb` file. For further details, we refer the reader to the accompanying Supplementary Note 2.

To overlay user data on the global network, first, user-provided gene expression, protein abundance and/or metabolite abundances' names are mapped to Metaboverse compatible identifiers. Metaboverse accepts any data input that can be appropriately mapped to a standard gene, protein or metabolite identifier or name. For components that Metaboverse is unable to map, a table is returned to the user so they can provide alternative names to aid in mapping. Second, provided data values are mapped to the appropriate nodes in the network. In cases where gene expression data are available, but protein abundance values are missing, Metaboverse will take the average of the available gene expression values to broadcast to the protein node. For complexes, the median of all available component values (metabolites, proteins and so on) is calculated (Supplementary Note 3, equation (12)). An aggregated $P$ value is inferred by multiplying the geometric mean of the $P$ values, as in refs. 57,58 (Supplementary Note 3, equation (13)).

### Collapsing reactions with missing values

After data mapping is complete, Metaboverse will generate a collapsed network representation for optional viewing during later visualization. Metaboverse enforces a limit of up to three reactions that can be collapsed as data down a pathway should only be inferred so far. Reaction collapsing allows for partial matches between inputs and outputs of two reactions to account for key metabolic pathways where a metabolite that is output by one reaction may not be required for the subsequent reaction. To perform a partial collapse, Metaboverse operates by largely the same scheme as outlined below, but additionally if a perfect match between reactions is not available, checks for partial matches by filtering out high-degree nodes (quartile 98 of all non-reaction node degrees) and then checking if, by default, at least 30% of the nodes match with its neighbour. For further details, we refer the reader to the accompanying Supplementary Note 2 and Extended Data Fig. 3.

### Regulatory pattern searches and prioritization

Metaboverse provides a variety of different regulatory patterns for the user to explore. To identify a reaction pattern is to compare some value that is computed from a reaction with a user-specified threshold. Equations for the reaction patterns available at the time of publication are included in the accompanying Supplementary Note 3 (equations (1)–(11)). Metaboverse provides a variety of reaction pattern sorting methods, which are detailed further in the accompanying Supplementary Notes 2 and 3. A complete and current list and description of all available reaction pattern modules can be found in the documentation at https://metaboverse.readthedocs.io.

Stoichiometry is not directly accounted for in reaction pattern identification as $\log_2$(fold change) values are utilized and factor out stoichiometric constants between experiment and control conditions and instead focus on relative magnitude changes of a given reaction component.

### Nearest neighbourhood searches and prioritization

To visualize all connections to a given network component, a user can select an entity (a gene, protein or metabolite) and visualize all reactions in which the component is involved. By doing so, the user can visualize other downstream effects the change of one entity might have across the total network, which consequently aids in bridging and identifying any reaction that may occur between canonically annotated pathways. These neighbourhoods can be expanded to view multiple downstream reaction steps and their accompanying genes, proteins and metabolites by modulating the appropriate user option in the software.

The user can also limit which entities are shown by enforcing a degree threshold. By setting this value at 50, for example, the network would not show nodes that have 50 or more connections. One caveat, however, is that this feature will occasionally break synchronous pathways into multiple pieces if one of these high-degree nodes formed the bridge between two ends of a pathway.

### Perturbation networks

Perturbation networks are generated by searching each reaction in the total reaction network for any reaction where at least one component is substantially perturbed. The user can modify the necessary criteria to base the search on the expression or abundance value or the statistical value and can choose the thresholding value to be used. For the expression thresholding, the provided value is assumed to be the absolute value, so a thresholding value of 3 would include any reactions where at least one component showed a greater than 3 measured change or less than −3 measured change, the value of which is dependent on the data provided by the user. Thus, these networks could represent reactions where a component was perturbed to a notable degree on a $\log_2$ fold change scale, $z$-score scale, or another appropriate unit for that biological context. Once a list of perturbed reactions is collected, the network is constructed, including each of these reactions and their components. Perturbed neighbouring reactions that share components are thus connected within the network, and perturbed reactions that are not next to other perturbed reactions are shown as disconnected subnetworks.

### Network visualization and exploration

Force-directed layouts of networks are constructed using D3 (https://d3js.org) by taking a user-selected pathway or entity and querying the reactions that are components of the selected pathway or entity. All inputs, outputs, modifiers and other components of these reactions, along with edges where both source and target are found in the subnetwork as nodes, are included and displayed. Relevant metadata, such as user-provided data and reaction descriptions, can be accessed by the user in real time. To visualize a pathway, a user selects a pathway, and all component reactions and their substrates, products, modifiers and metadata are queried from the total reaction database. Super-pathways help categorize these pathways and are defined as any pathway containing more than 200 nodes.

Time course and multiple condition experiments are automatically detected from the user's input data. When users provide these data and specify the appropriate experimental parameters on the variable input page, they will have the option to provide timepoint or condition labels. Provided data should be listed in the data table in the same order that the labels are provided. Within all visualization modules, the data for each timepoint or condition can then be displayed using a slider bar, which will allow the user to cycle between timepoints or conditions.

Compartments are derived from Reactome annotations[17–19]. Compartment visualizations are generated using D3's hull plotting feature. Compartment boundaries are defined at the reaction levels and made

to encompass each reaction's substrates, products and modifiers for that given compartment.

Some performance optimization features are included by default to prevent computational overload. For example, nearest neighbour subnetworks with more than 1,500 nodes, or nodes with more than 500 edges, will not be plotted because the plotting of this information in real time can be prohibitively slow.

## Software packaging

The Metaboverse application is packaged using Electron (https://electronjs.org). Back-end network curation and data processing are performed using Python (https://www.python.org/) and the NetworkX[59], pandas[60,61], NumPy[62], SciPy[61,63] and Matplotlib[63] libraries. This back-end functionality is packaged as a single, operating system-specific executable using the PyInstaller library (https://www.pyinstaller.org) and is available to the app's visual interface for data processing. Front-end visualization is performed using Javascript and relies on the D3 (https://d3js.org) and JQuery packages (https://jquery.com). Saving network representations to a PNG file is performed using the (https://github.com/edeno/d3-save-svg) and string-pixel-width (https://github.com/adambisek/string-pixel-width) packages. Documentation for Metaboverse is available at https://metaboverse.readthedocs.io. Continuous integration services are performed by GitHub Actions to routinely run test cases for each change made to the Metaboverse architecture. The Metaboverse source code can be accessed at https://github.com/Metaboverse/metaboverse. The code used to draft and revise this manuscript, as well as all associated scripts used to generate and visualize the data presented in this manuscript, can be accessed at ref. 55.

## Human LUAD metabolomics and analysis

Data were accessed from Metabolomics Workbench project PR000305 and processed as in our previous re-study of these data[24]. P values were derived using a two-tailed, homoscedastic Student's t-test and adjusted using the Benjamini–Hochberg correction procedure.

The initial Kaplan–Meier survival analysis was performed using tools and data hosted on The Human Protein Atlas (version 20.1; released 24 February 2021) (refs. 65–67). Survival analysis as displayed in this manuscript was performed in R (version 4.0.3) using the survival (version 3.2-11) and survminer (version 0.4.9) packages. Correlation analysis between Metaboverse reaction pattern rank and survival statistic was performed using the Pearson correlation coefficient and a loess regression using the ggscatter() function from the ggpubr (version 0.4.0) package. TCGA FPKM gene expression data were obtained from the Human Protein Atlas project (https://www.proteinatlas.org/download/rna_cancer_sample.tsv.zip) and clinical patient data were obtained from TCGA (https://portal.gdc.cancer.gov/projects/TCGA-LUAD). Clinical data were censored as 'Dead' or 'Alive', and 'Alive' patients were right censored using days since last follow-up. Patients were stratified into two gene expression groups (High, Low) using the optimized surv_cutpoint() function from the survminer package (version 0.4.9) with the minimum proportion for a group set at 0.2 (ref. 33).

DepMap data (21Q4 Public) were subsetted to include only non-small cell lung cancer cell lines. The scatter plot was generated from the DepMap online interface (https://depmap.org/).

## Yeast growth assays and experiments

The S. cerevisiae BY4743 (MATa/α, his3/his3, leu2/leu2, ura3/ura3, met15/MET15 and lys2/LYS2) WT or mct1Δ strains as described in ref. 40 were used for all yeast experiments. Growth assays were performed using S-minimal (S-min) medium with no uracil added and containing either 2% glucose or 2% raffinose. Equal numbers of WT or mct1Δ yeast transformed with empty vector (EV), CTP1 overexpression and CTP1-C-terminal GFP-overexpression plasmids were spotted as tenfold

serial dilutions during mid-log phase ($OD_{600}$ 0.3–0.6). Plates were incubated at 30 °C for 2–3 days before imaging.

## Protein expression validation

Yeast cultures were grown to mid-log phase and lysed. Proteins were run on a sodium dodecly-sulfate-page gel and assayed using antibodies for α-GFP (1/2,000 dilution; rabbit; Cell Signaling Technology no. 2956; RRID: AB_1196615) and α-Pgk1 (1/3,000 dilution; mouse; Abcam no. ab113687; RRID: AB_10861977).

## RNA-seq

RNA-seq data were generated by growing S. cerevisiae biological replicates for strains mct1Δ (n = 4) and WT (n = 4). Briefly, cells were grown in glucose and switched to raffinose-supplemented growth medium (SR-URA) for 0, 3 and 12 h, such that at the time of collection, cultures were at $OD_{600}$ of 1. Cultures were flash frozen, and later total RNA was isolated using the Direct-zol kit (Zymo Research) with on-column DNase digestion and water elution. Sequencing libraries were prepared by purifying intact poly(A) RNA from total RNA samples (100–500 ng) with oligo(dT) magnetic beads, and stranded messenger RNA-seq libraries were prepared as described using the Illumina TruSeq Stranded mRNA Library Preparation Kit (RS-122-2101 and RS-122-2102). Purified libraries were qualified on an Agilent Technologies 2200 TapeStation using a D1000 ScreenTape assay (cat. nos. 5067-5582 and 5067-5583). The molarity of adaptor-modified molecules was defined by quantitative polymerase chain reaction (PCR) using the Kapa Biosystems Kapa Library Quant Kit (cat. no. KK4824). Individual libraries were normalized to 5 nM, and equal volumes were pooled in preparation for Illumina sequence analysis. Sequencing libraries (25 pM) were chemically denatured and applied to an Illumina HiSeq v4 single-read flow cell using an Illumina cBot. Hybridized molecules were clonally amplified and annealed to sequencing primers with reagents from an Illumina HiSeq SR Cluster Kit v4-cBot (GD-401-4001). Following the transfer of the flow cell to an Illumina HiSeq 2500 instrument (HCSv2.2.38 and RTA v1.18.61), a 50-cycle single-read sequence run was performed using HiSeq SBS Kit v4 sequencing reagents (FC-401-4002).

Sequence FASTQ files were processed using XPRESSpipe (version 0.6.0) (ref. 68). Batch and log files are available at ref. 55. Notably, reads were trimmed of adaptors (AGATCGGAAGAGCACACGTCTGAACTC-CAGTCA). On the basis of library complexity quality control, de-duplicated alignments were used for read quantification due to the high number of duplicated sequences in each library. Differential expression analysis was performed using DESeq2 (version 1.22.1) (ref. 69) by comparing mct1Δ samples with WT samples at the 12 h timepoint to match the steady-state proteomics data. $\log_2$(fold change) and false discovery rate (FDR; 'p-adj') values were extracted from the DESeq2 output.

## Proteomics

Steady-state quantitative proteomics data were previously processed and obtained from ref. 40 (ProteomeXchange: PXD035000).

Briefly, cells were grown in glucose and switched to SR-URA overnight and collected at the mid-log phase. Cells were resuspended, lysed and clarified. Proteins were then subjected to disulfide reduction and digested for 16 h with LysC (1:100 enzyme:protein ratio) at room temperature, followed by trypsin (1:100 enzyme:protein ratio) for 6 h at 37 °C. Proteins were again quantified and subjected to tandem mass tag (TMT)-11 labelling, after which samples were pooled. Pooled TMT-labelled peptide samples were fractionated using basic pH reversed-phase high-performance liquid chromatography (LC) (Agilent 1260 Infinity pump equipped with a degasser and a single wavelength detector set at 220 nm). Fractions were desalted via StageTip, dried via vacuum centrifugation and reconstituted in 5% acetonitrile, 5% formic acid for LC–tandem mass spectrometry (MS/MS) processing. MS data were collected using an Orbitrap Fusion Lumos mass

spectrometer (Thermo Fisher Scientific) equipped with a Proxeon EASY-nLC 1000 LC system (Thermo Fisher Scientific). The multi-notch MS3-based TMT method was used[70]. MS2 mass spectra were processed using the Sequest algorithm[71]. Spectra were converted to mzXML, followed by database searching using the yeast proteome downloaded from Uniprot (UniProt-Consortium, 2015) in both forward and reverse directions, along with common contaminating protein sequences. Peptide-spectrum matches were adjusted to a 1% FDR[72]. Linear discriminant analysis was used to filter peptide-spectrum matches, as described previously[73]. Each TMT channel was summed across all quantified proteins and normalized to enforce equal protein loading. Each protein's quantitative measurement was then scaled to 100.

For the analysis used within this manuscript, we compared the $mct1\Delta$ ($n = 3$) with the WT ($n = 3$) cell populations. $\log_2$(fold change) values and Benjamini–Hochberg-corrected $P$ values were generated by comparing $mct1\Delta$ with the WT cells. $P$ values were generated before correction using a two-tailed, homoscedastic Student's $t$-test.

### Gas chromatography metabolomics

Metabolomics data were generated by growing the appropriate yeast strains in synthetic complete media supplemented with 2% glucose until they reached saturation ($n = 6$; except in one 3 h WT sample, where $n = 5$). Cells were then transferred to S-minimal medium containing 2% raffinose and leucine and collected after 0, 15, 30, 60 and 180 min ($n = 6$ per timepoint per strain, except for the 3 h WT samples, where $n = 5$) at $OD_{600}$ 0.6–0.8.

A 75% boiling ethanol (EtOH) solution containing the internal standard d4-succinic acid (Sigma 293075) was then added to each sample. Boiling samples were vortexed and incubated at 90 °C for 5 min. Samples were then incubated at −20 °C for 1 h. After incubation, samples were centrifuged at 5,000$g$ for 10 min at 4 °C. The supernatant was then transferred from each sample tube into a labelled, fresh 13 × 100 mm glass culture tube. A second standard was then added (d27-myristic acid CDN Isotopes: D-1711). Pooled quality control samples were made by removing a fraction of the collected supernatant from each sample, and process blanks were made using only extraction solvent and no cell culture. The samples were then dried en vacuo. This process was completed in three separate batches.

All gas chromatography–MS analysis was performed with an Agilent 5977b GC-MS MSD-HES and an Agilent 7693A automatic liquid sampler. Dried samples were suspended in 40 µl of 40 mg ml$^{-1}$ $O$-methoxyamine hydrochloride (MP Bio no. 155405) in dry pyridine (EMD Millipore no. PX2012-7) and incubated for 1 h at 37 °C in a sand bath. Then, 25 µl of this solution was added to autosampler vials and 60 µl of $N$-methyl-$N$-trimethylsilyltrifluoroacetamide (MSTFA) with 1% trimethylchlorosilane (TMCS) (Thermo Fisher no. TS48913) was added automatically via the autosampler and incubated for 30 min at 37 °C. After incubation, samples were vortexed, and 1 µl of the prepared sample was injected into the gas chromatograph inlet in the split mode with the inlet temperature held at 250 °C. A 10:1 split ratio was used for the analysis of the majority of metabolites. For those metabolites that saturated the instrument at the 10:1 split concentration, a split of 50:1 was used for the analysis. The gas chromatograph had an initial temperature of 60 °C for 1 min followed by a 10 °C min$^{-1}$ ramp to 325 °C and a hold time of 5 min. A 30 m Phenomenex Zebron AB-5HT with a 5 m inert Guardian capillary column was employed for chromatographic separation. Helium was used as the carrier gas at a rate of 1 ml min$^{-1}$.

Data were collected using MassHunter software (Agilent). Metabolites were identified, and their peak area was recorded using MassHunter Quant. These data were transferred to an Excel spreadsheet (Microsoft). Metabolite identity was established using a combination of an in-house metabolite library developed using pure purchased standards, and the NIST (https://www.nist.gov) and Fiehn libraries[74]. Resulting data from all samples were normalized to the internal standard d4-succinate. $P$ values were derived using a homoscedastic, two-tailed

Student's $t$-test and adjusted using the Benjamini–Hochberg correction procedure.

### Liquid chromatography metabolomics

Metabolomics data were generated by growing the appropriate yeast strains in synthetic complete medium supplemented with 2% glucose until they reached saturation ($n = 3$). Cells were then transferred to S-minimal medium containing 2% raffinose and leucine and collected after approximately 8 h ($n = 3$) at $OD_{600}$ 0.6–0.8.

The procedures for metabolite extraction were performed as previously described[75]. Yeast cultures were pelleted, snap-frozen and kept at −80 °C. Then 5 ml of 75% boiled EtOH was added to every frozen pellet. Pellets were vortexed and incubated at 90 °C for 5 min. All samples were then centrifuged at 5,000$g$ for 10 min. Supernatants were transferred to fresh tubes, evaporated overnight in a Speed Vacuum and then stored at −80 °C until they were run on the mass spectrometer.

The conditions for LC are described in previous studies[76,77]. Briefly, a hydrophilic interaction LC method with an Xbridge amide column (100 × 2.1 mm, 3.5 µm) (Waters) was employed on a Dionex (Ultimate 3000 UHPLC) for compound separation and detection at room temperature. The mobile phase A was 20 mM ammonium acetate and 15 mM ammonium hydroxide in water with 3% acetonitrile, pH 9.0, and the mobile phase B was acetonitrile. The linear gradient was as follows: 0 min, 85% B; 1.5 min, 85% B, 5.5 min, 35% B; 10 min, 35% B, 10.5 min, 35% B, 14.5 min, 35% B, 15 min, 85% B and 20 min, 85% B. The flow rate was 0.15 ml min$^{-1}$ from 0 to 10 min and 15 to 20 min, and 0.3 ml min$^{-1}$ from 10.5 to 14.5 min. All solvents were LC–MS grade and purchased from Thermo Fisher Scientific.

MS was performed as described in previous studies[76,77]. Briefly, the Q Exactive MS (Thermo Scientific) is equipped with a heated electrospray ionization probe, and the relevant parameters are as listed: evaporation temperature, 120 °C; sheath gas, 30; auxiliary gas, 10; sweep gas, 3; spray voltage, 3.6 kV for positive mode and 2.5 kV for negative mode. The capillary temperature was set at 320 °C and S-lens was 55. A full scan range from 60 to 900 m/z was used. The resolution was set at 70,000. The maximum injection time was 200 ms. Automated gain control was targeted at 3,000,000 ions.

Data were collected, metabolites were identified and their peak area was recorded using El-Maven software (version 0.12.0) (refs. 78–80). A pre-entered compound list of m/z values and corresponding metabolites was utilized to enable El-Maven EIC (extracted ion chromatogram) extraction of all samples. A manual visual examination of peaks selected by El-Maven was performed and misannotated peaks were manually corrected and exported as an Excel spreadsheet (Microsoft), as described in refs. 78,80. Metabolite identity was established using a combination of an in-house metabolite library developed using pure purchased standards, and the NIST (https://www.nist.gov) and Fiehn libraries[74]. $P$ values were derived using a homoscedastic, two-tailed Student's $t$-test and adjusted using the Benjamini–Hochberg correction procedure.

### Correlation analysis

To correct the expression bias arising from highly expressed genes, gene expression data were first corrected using spatial quantile normalization (SpQN; version 1.0.0) for each dataset with the first four principal components being removed for each dataset[50]. Genes were considered co-expressed in refine.bio datasets if SpQN-normalized Pearson's $r > 0.5$ and in the WT data generated for this study if >0.75.

Gene Ontology (GO) enrichment analysis was performed by processing the correlated gene sets from each dataset using the PANTHER Overrepresentation Test (version 16; released 20210224) on the GO biological process complete annotation dataset (https://doi.org/10.5281/zenodo.4735677; released 2021-05-01) (refs. 81,82) via the GO Resource[83,84]. Enrichments were determined using Fisher's exact test and $P$ values were corrected using the PANTHER FDR calculation[81,82]. Enrichments were prioritized by fold change. For overlapping

GO terms, the GO term with the highest fold change was used for the visualization. Enrichment FDRs and fold changes were visualized as bubble plots generated using seaborn (version 0.11.0) and Matplotlib (version 3.4.2) (refs. [63],[85]). Scatter plots of co-expressed genes against the gene of interest were generated using the `regplot()` function from seaborn (version 0.11.0) and Matplotlib (version 3.4.2) (refs. [63],[85]).

### Sensitivity analysis

For the original input metabolomics datasets from Wikoff, et al.[29] and the 12 h timepoint *mct1*Δ yeast, six replicates each of 10%, 20%, 30%, 40%, 50%, 60%, 70%, 80% and 90% of metabolites missing were generated. For the proteomics dataset for the 12 h timepoint *mct1*Δ yeast, 0%, 30% or 60% of proteins missing were generated as above. The random seed for each replicate was distinct but consistent between re-analyses. Each of these datasets were then processed by Metaboverse version 0.10.0 using default parameters, and 'Average' patterns from the Wikoff dataset and 'ModReg' patterns from the *mct1*Δ were output. Box plots and heat maps of the resulting data were generated using pandas (version 1.4.0) (refs. [59],[60]), NumPy (version 1.22.3) (ref. [61]), Matplotlib (version 3.5.1) (ref. [63]) and Seaborn (version 0.12.1) (ref. [85]). Quantification of the number of reactions that could be measured with or without reaction collapsing was performed by counting the number of reactions with at least two reaction component types (reactants, products and modifiers) each having at least one measurement. Scripts to reproduce these analyses are available at ref. [64].

### Other visualization

Heat maps were generated using the `clustermap()` function from seaborn (version 0.11.0) and Matplotlib (version 3.4.2) using custom gene, protein or metabolite lists[63],[85]. Heat map values were mean centred at 0 (*z* score). Hierarchical clustering was performed where indicated by the linkage lines using a simple agglomerative (bottom-up) hierarchical clustering method (or unweighted pair group method with arithmetic mean (UPGMA)). Gene comparison box plots were generated using the `swarmplot()` and `boxplot()` functions from seaborn (version 0.11.0) and Matplotlib (version 3.4.2) (refs. [63],[85]). Single-cell dot plots were generated using scanpy (version 1.8.2) (ref. [86]). The pathway images in Fig. 4d and Supplementary Fig. 7 were generated manually using Adobe Illustrator (https://www.adobe.com/). Hex values for each protein or metabolite were determined by converting the fold change values in Python https://www.python.org/) using Matplotlib[64].

### Biological materials availability

All biological materials related to this manuscript are available without restriction. Requests for biological materials should be directed to J.R. (rutter@biochem.utah.edu).

### Statistics and reproducibility

*P* values associated with log$_2$ fold changes for Metaboverse integration were calculated using a two-tailed, homoscedastic Student's *t*-test and adjusted using the Benjamini–Hochberg correction procedure, except for RNA-seq data, which used DESeq2 (version 1.22.1) to determine FDRs[69]. Other details are listed in Methods. For yeast experiments, samples were prepared with separate and fresh preparations with three to six biological replicates in each experimental or control group, as detailed in Methods and elsewhere as appropriate within the manuscript. In the case of the refine.bio yeast cohort, the entire WT sample cohort was used as specified in the manuscript text. For the public human LUAD datasets, the Wikoff 2015 study[29] contained 39 tumour tissue samples and 39 paired normal tissue samples; and TCGA data contained 487 gene expression samples total that were relevant to this study. No statistical method was used to predetermine the sample size. Sample sizes for high-throughput data generated for this study were chosen on the basis of a first-principles estimated understanding of the number of samples needed to generate expected statistical

distributions on the basis of the data type. Statistical values were then adjusted for false positives following the convention for the respective data type. Other data were previously generated for other studies. For survival analysis, TCGA data were right censored and then removed if no days to death or censored days to death were available. Metabolomics samples that did not pass basic quality control (*n* = 1) were excluded from further analysis. No additional data were excluded. All biological assays were repeated at least three times. All replication attempts were successful. Verification of plasmid construct expression by western blot was performed once as a simple validation that the construct was being overexpressed. Samples were randomized during sample preparation, but not during sample collection for the yeast RNA-seq and yeast metabolomics data, or were previously collected (human TCGA data, human metabolomics data and yeast proteomics data), or otherwise not amenable to randomization (yeast growth spot tests and so on). Yeast sample ordering and handling would have otherwise been randomized during sample processing. Data were either previously collected (human TCGA data, human metabolomics data and yeast proteomics data), blinded during sample preparation but not sample collection (yeast RNA-seq and yeast metabolomics) or were otherwise not amenable to blinding (yeast growth spot tests and so on). Yeast samples were additionally difficult to blind during growth and collection as exact growth rates need to be measured throughout, and often correlate with genetic background.

All analysis code, raw and processed data are available in the above-named repositories or at ref. [55]. This archive is organized by the figure number, and the source data, references to source data and code needed to replicate associated figures are included.

### Protocols

A step-by-step protocol describing use of Metaboverse can be found at Nature Protocol Exchange[87].

### Reporting summary

Further information on research design is available in the Nature Portfolio Reporting Summary linked to this article.

## Data availability

Gene expression counts for LUAD were obtained from the Human Protein Atlas project's TCGA FPKM gene expression data (https://www.proteinatlas.org/download/rna_cancer_sample.tsv.zip) and clinical patient data were obtained from TCGA (https://portal.gdc.cancer.gov/projects/TCGA-LUAD). Single-cell data were obtained from the Human Lung Cell Atlas project version 1.0 (https://zenodo.org/record/6337966#.YkzVrOjMIQ-). Metabolomics data were obtained from the Metabolomics Workbench repository under project identifier PR000305, study identifiers ST000390 and ST000391. *mct1*Δ and accompanying WT transcriptomics time course data are deposited at the GEO repository under identifier GSE151606. *mct1*Δ and WT proteomics data are deposited at the ProteomeXchange repository under identifier PXD035000. Metabolomics data are deposited at the Metabolomics Workbench repository under project identifier PR000961, study identifier ST001401 and project identifier PR001422, study identifier ST002232. For gene co-expression analyses, all yeast samples available at https://www.refine.bio were accessed and downloaded on 16 March 2021 (ref. [49]). The curated Metaboverse files for the datasets analysed for this manuscript were processed using Metaboverse version 0.9.0, unless otherwise specified, and are available at ref. [55]. Source data are provided with this paper. All other source data needed to replicate the results of this work have been deposited in the above-mentioned repositories or are available at ref. [55].

## Code availability

The Metaboverse source code is available at https://github.com/Metaboverse/Metaboverse and https://github.com/Metaboverse/

metaboverse-cli. The latest version of the software can be found at https://github.com/Metaboverse/Metaboverse/releases/latest. The source code and data for this manuscript and the related analyses are available at ref. 55. Source code is archived at Zenodo: https://zenodo.org/record/7384508 and https://zenodo.org/record/7384509.

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

## Acknowledgements

We thank T. O'Connell, P. Trébulle and J. Rabinowitz for valuable feedback and feature suggestions during the beta-testing of the software. We thank A. Bott, S. Fogarty, K. Hicks, J. Morgan and other members of the Rutter lab for their thoughtful insights, suggestions and critical feedback during the drafting of the manuscript. We thank J. Van Vranken for assistance in depositing the proteomics data. We thank B. Dalley and the University of Utah High-Throughput Genomics Core for help with RNA library preparation and sequencing. We thank G. Lam for assistance in generating the CTP1–GFP-OE plasmid. We also thank C. Pickett for providing editing services for this manuscript. The support and resources from the Center for High-Performance Computing at the University of Utah are gratefully acknowledged. J.A.B. is supported by the National Cancer Institute (NCI) of the National Institutes of Health (NIH) under award number F99CA253744 (https://www.cancer.gov) and received additional support from the National Institute of Diabetes and Digestive and Kidney Diseases (NIDDK) Inter-disciplinary Training Grant T32 Program in Computational Approaches to Diabetes and Metabolism Research, T32DK11096601 to W.W. Chapman and S.J. Fisher (https://www.niddk.nih.gov). S.M.N. received support from The United Mitochondrial Disease Foundation PF-15-046 (https://www.umdf.org) and the American Cancer Society PF-18-106-01 (https://www.cancer.org) postdoctoral fellowships, along with T32HL007576. J.E.C. is funded by S10OD016232, S10OD021505 and U54DK110858. This work was

supported by funds from the NIDDK fellowship 1T32DK11096601 and the NCI fellowship 1F99CA253744 (to J.A.B.) (https://www.niddk.nih.gov, https://www.cancer.gov), the NSF grants DBI-1661375 and IIS-1513616 (to B.W.) (https://www.nsf.gov), and the NIH grant R35GM131854 (to J.R.) (https://www.nih.gov). The computational resources used were partially funded by the NIH Shared Instrumentation Grant 1S10OD021644-01A1 (https://www.nih.gov). MS equipment was obtained through NCRR Shared Instrumentation Grant 1S10OD016232-01, 1S10OD018210-01A1 and 1S10OD021505-01 (to J.E.C.). The funders had no role in study design, data collection and analysis, decision to publish or preparation of the manuscript. The content of this manuscript is solely the responsibility of the authors and does not necessarily represent the official views of the NIH.

## Author contributions

Y.Z. and Y.O. contributed equally to this work. Conceptualization: J.A.B., T.C.W., B.W. and J.R. Supervision: J.A.B., B.W. and J.R. Project administration: J.A.B. Investigation: J.A.B., Y.O., T.C.W., A.A.C., M.E.C., S.M.N. and T.V.R. Formal analysis: J.A.B., Y.Z., Y.O., A.A.C. and T.V.R. Software: J.A.B., Y.Z. and I.G. Methodology: J.A.B. and Y.Z. Validation: J.A.B., Y.O., A.A.C., M.E.C. and I.G. Data curation: J.A.B., Y.O., T.C.W., A.A.C., S.M.N. and T.V.R. Resources: J.A.B., J.E.C., B.W. and J.R. Funding acquisition: J.A.B., J.E.C., B.W. and J.R. Writing—original draft preparation: J.A.B. Writing—review and editing: J.A.B., Y.Z., Y.O., T.C.W., A.A.C., T.V.R, I.G., S.M.N., J.E.C., B.W. and J.R. Visualization: J.A.B., Y.Z. and Y.O.

## Competing interests

The authors declare no competing interests.

## Additional information

**Extended data** is available for this paper at https://doi.org/10.1038/s41556-023-01117-9.

**Correspondence and requests for materials** should be addressed to Jordan A. Berg or Jared Rutter.

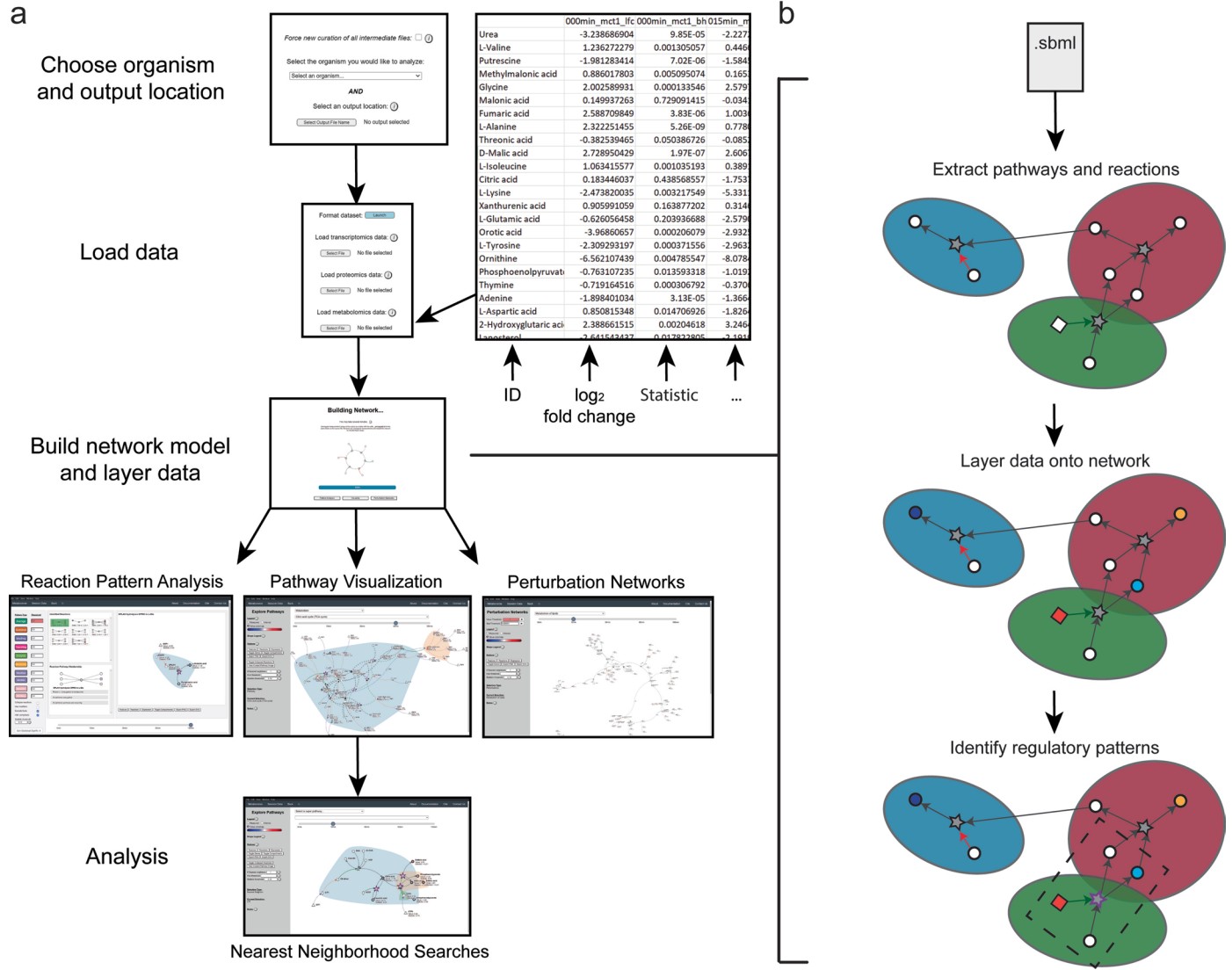

**Extended Data Fig. 1 | Metaboverse provides a simple, interactive user interface for processing and exploring multi-omics datasets and metabolic patterns. a**. An overview of the graphical user interface of Metaboverse and a summary of analytical submodules contained within the platform. **b**. Overview of back-end metabolic network curation and data layering. .sbml refers to the systems biology markdown language-formatted resource containing the metabolic network information for the model organism. Colored circles represent subcellular compartments, biochemical reactions are stars, metabolites are circles, and proteins are squares. Grey arrows represent core reaction relationships and red arrows represent reaction inhibitor relationships. Progressively blue shades indicate decreased measurements between two conditions and progressively red shades indicate increased measurements between two conditions. The dashed box represents an identified reaction pattern, or net change across the measured components of a reaction.

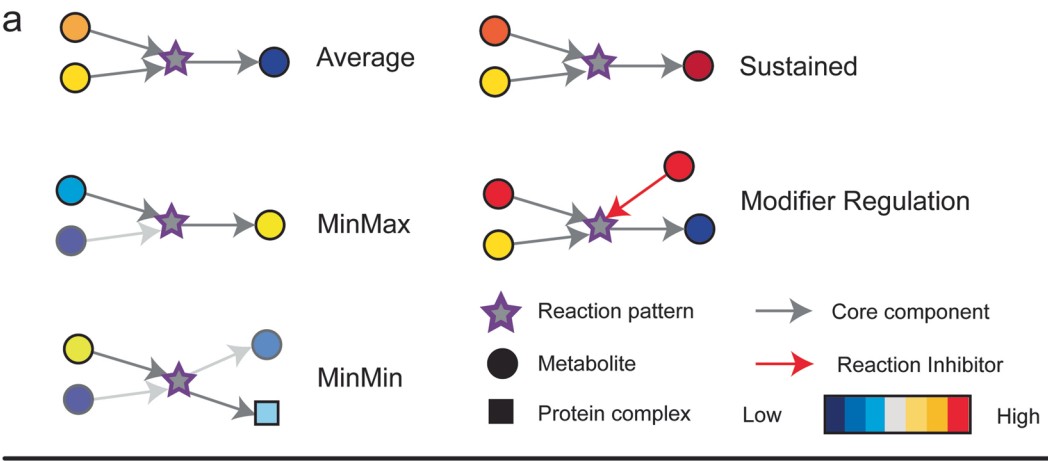

**Extended Data Fig. 2 | Metaboverse identifies a variety of metabolic patterns and enables pattern identification with sparse measurements. a**. Examples of a selection of reaction patterns available in Metaboverse. Reactions are depicted as stars, metabolites as circles, protein complexes as squares, and proteins as diamonds. Core interactions (inputs, outputs) are depicted as grey arrows, reaction catalysts as green arrows, and reaction inhibitors as red arrows. Component measurements are depicted in a blue-to-red color map, where lower values are more blue and higher values are more red. **b**. Example sub-networks where a reaction collapse would occur. Measured components are depicted as red circles, unmeasured components as white circles, and reactions as stars. Core interactions (inputs, outputs) are depicted as grey lines and identical components that would form the bridge between two reactions are depicted as dashed black lines between circles. A collapsed reaction is depicted as a star with a dashed border and its summarized connections between measured components are dashed black lines between a measured component and a reaction node.

For each reaction in the network:
 If at least one component on each side of the reaction is measured:
 Keep reaction in network:

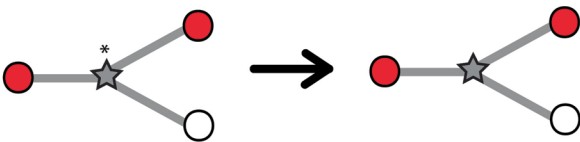

 Else if both sides not measured:
 Search for neighboring reactions with a measurement on opposing side:
 If an input neighbor and an output neighbor both have a measurement,
 generate a collapsed reaction of the three reactions:

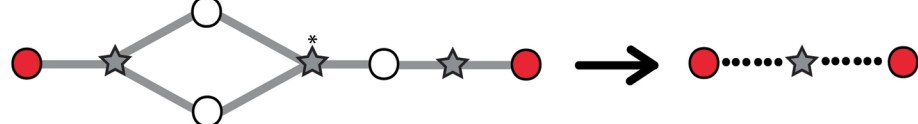

 Else if one side is measured and the other is not:
 Search for neighboring reactions for the side of the reaction with missing
 measurements:
 If the neighboring reaction has a measurement, generate a collapsed
 reaction of the two reactions:

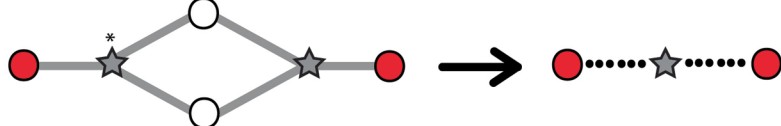

 Else:
 Keep reaction in network as is

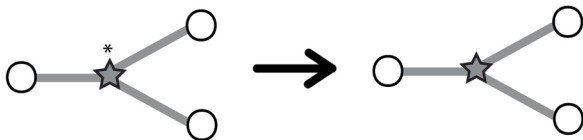

 Notes:
 - For a side of a reaction to be "measured," only one component needs to be measured:

 - For a reaction to be considered a "neighbor," only a percentage of the reactants/products
 need to match - default is 30% of non-hub nodes:

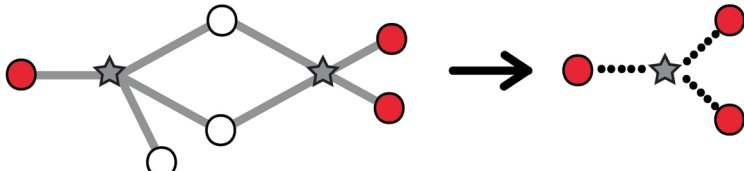

**Extended Data Fig. 3 | Visual summary of Algorithm 4.** Reactions are indicated by stars, and metabolites by circles. Measured reaction components are indicated by filled red circles, and unmeasured reaction components are indicated by empty white circles. The original reaction being considered at each step is marked with an asterisk. Collapsed edges are marked with dashed edges, and canonical reaction relationships are marked with solid edges.

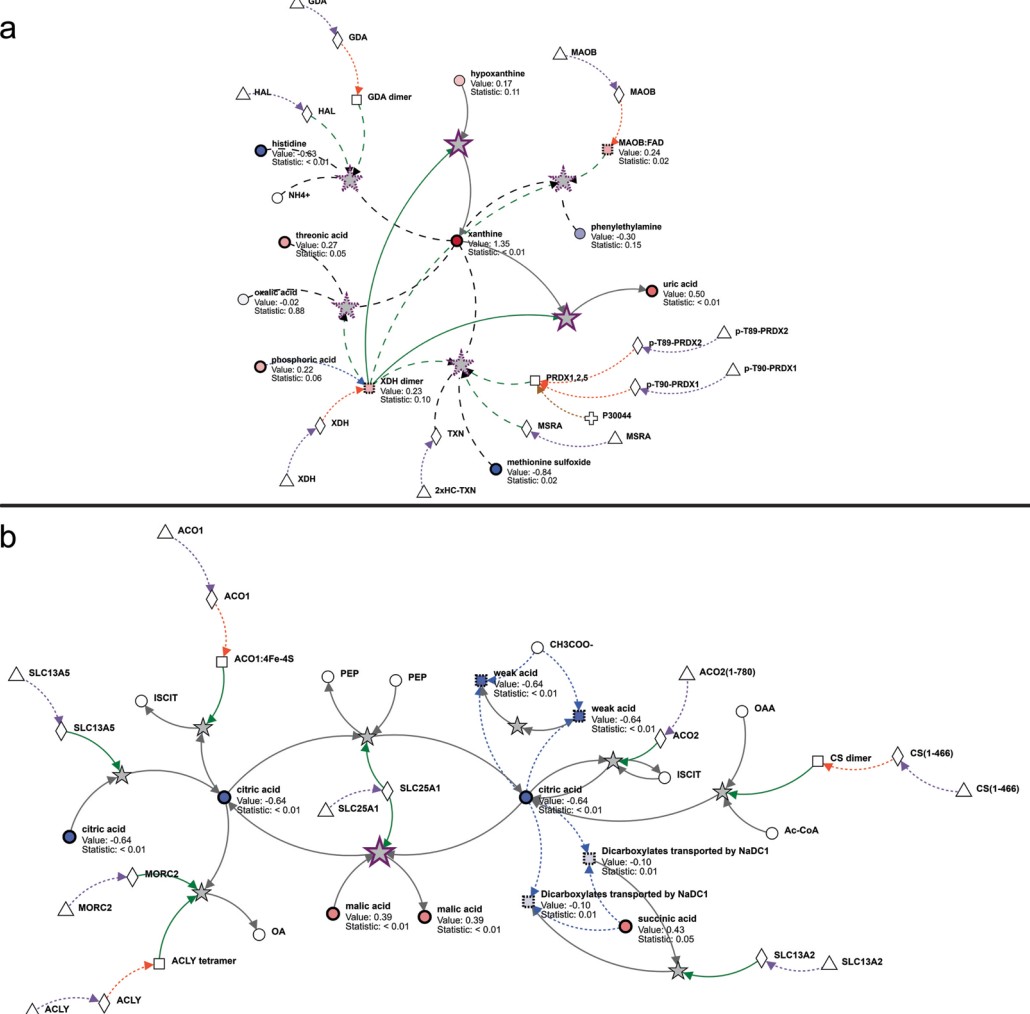

**Extended Data Fig. 4 | Metaboverse identifies reaction patterns in xanthine and TCA metabolism. a**. Identification of xanthine regulation by both the pattern recognition and perturbation analysis modules. **b**. Disruptions of TCA metabolism support canonical disruptions during adenocarcinoma development. Metabolomics values are shown as node shading, where an increasingly blue shade indicates downregulation, and an increasingly red shade indicates upregulation. Measured log$_2$(fold change) and statistical values for each entity are displayed below the node name. A gray node indicates a reaction. A bold gray node with a purple border indicates a motif at this reaction. Circles indicate metabolites, squares indicate complexes, and diamonds indicate proteins. Gray edges indicate core relationships between reaction inputs and outputs. Green edges indicate a catalyst, and red edges indicate inhibitors. Dashed blue edges point from a metabolite component to the protein complex in which it is involved. Dashed orange edges point from a protein component to the protein complex in which it is involved. Protein complexes with dashed borders indicate that the values displayed on that node were inferred from the constituent protein and metabolite measurements. Hub limit was set at 30 during generation of the network visualization as shown in sub-panel **b**. Source numerical data are available at[64].

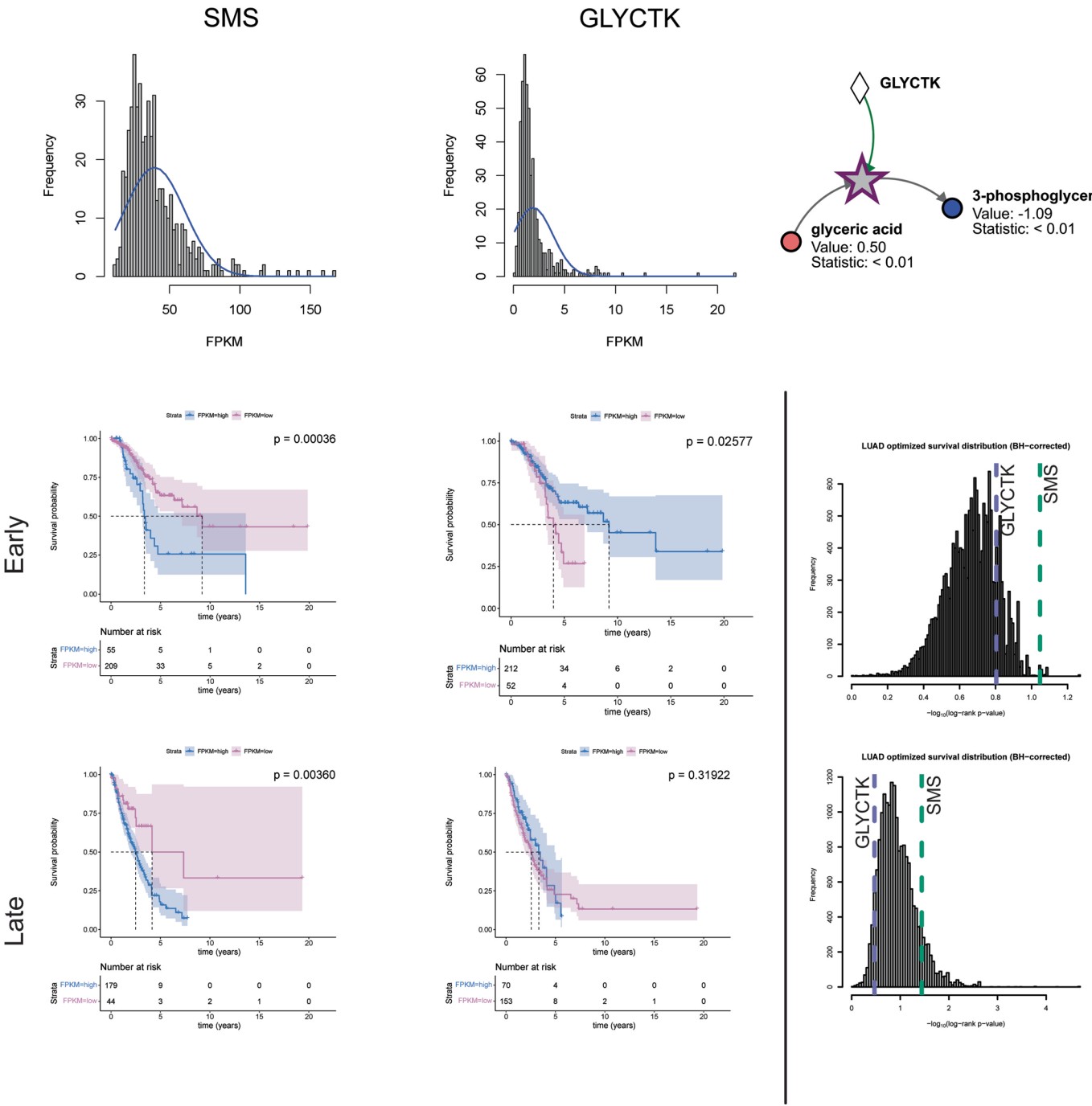

**Extended Data Fig. 5 | Overall survival outcomes correlations of *SMS* and *GLYCTK* gene expression in early-stage adenocarinomas are stronger than in later stage adenocarcinomas.** (top) gene FPKM distributions for *SMS* and *GLYCTK* (Glycerate Kinase, FPKM cut-off: 0.913; high: 104 tumors, low: 383 tumors). (middle) Kaplan-Meier plots using Cox regression analysis for early-stage (stage IA-B) samples for *SMS* and *GLYCTK* and distribution of all genes' Benjamini-Hochberg log-rank p-values. (bottom) Kaplan-Meier plots using

Cox regression analysis for late stage (stage II+) samples for *SMS* and *GLYCTK* and distribution of all genes' Benjamini-Hochberg log-rank p-values. Shading in Kaplain-Meier plots indicates 95% confidence intervals for each expression group. Dashed lines indicate median survival times for each expression group. Risk tables are displayed below each Kaplan-Meier plot, and include the number of individuals in each risk category at time = 0 years. Source numerical data are available[64].

a

## DepMap Dependency

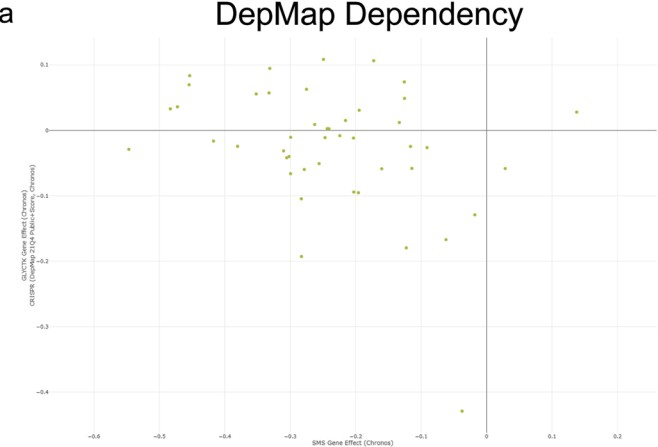

b

## Rank vs. Survival BH Value

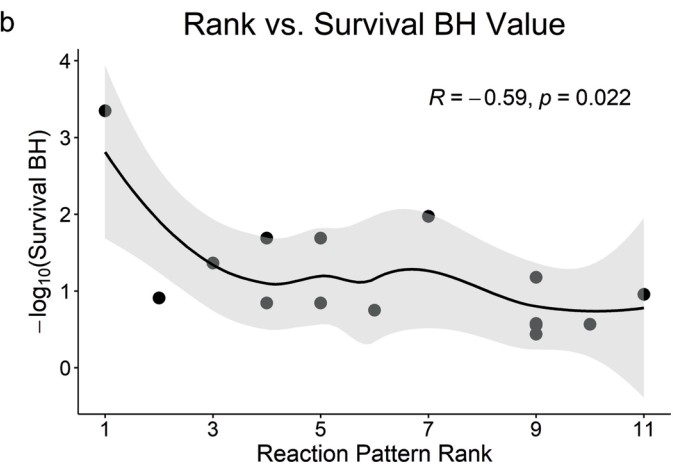

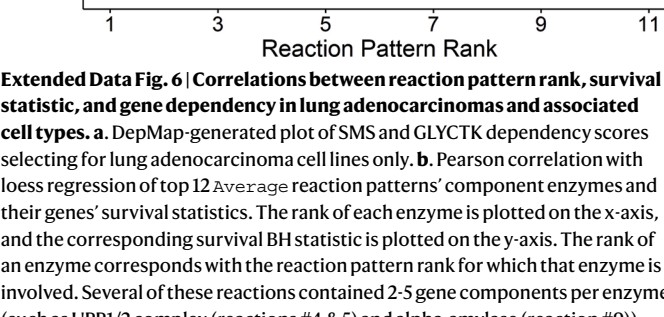

c

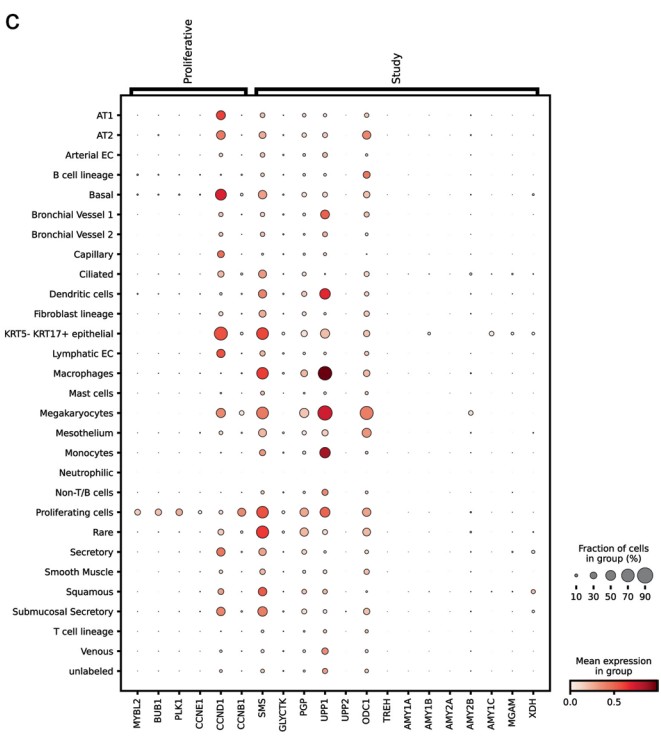

**Extended Data Fig. 6 | Correlations between reaction pattern rank, survival statistic, and gene dependency in lung adenocarcinomas and associated cell types. a**. DepMap-generated plot of SMS and GLYCTK dependency scores selecting for lung adenocarcinoma cell lines only. **b**. Pearson correlation with loess regression of top 12 Average reaction patterns' component enzymes and their genes' survival statistics. The rank of each enzyme is plotted on the x-axis, and the corresponding survival BH statistic is plotted on the y-axis. The rank of an enzyme corresponds with the reaction pattern rank for which that enzyme is involved. Several of these reactions contained 2-5 gene components per enzyme (such as UPP1/2 complex (reactions #4 & 5) and alpha-amylase (reaction #9)). Reactions without a rank (reaction #8, sucrase-isomaltase dimer) did not have

a corresponding survival statistic from the TCGA gene expression dataset to investigate. Shading indicates 95% confidence interval of regression. **c**. Dotplot of gene markers (horizontal axis) across annotated cell types (vertical axis) in the Human Lung Cell Atlas (HLCA) project. Proliferative gene markers were sourced from Whitfield, M., George, L., Grant, G. et al. Common markers of proliferation. Nat Rev Cancer 6, 99-106 (2006). The other genes displayed are based on the top Metaboverse hits' corresponding enzymes from the LUAD metabolomics dataset. Dot size is scaled by fraction of cells in a cell type group expressing the given gene marker. Dot shade is scaled by mean expression of gene marker within the cell type group. Source numerical data are available[64].

## Top ModReg Reaction Patterns

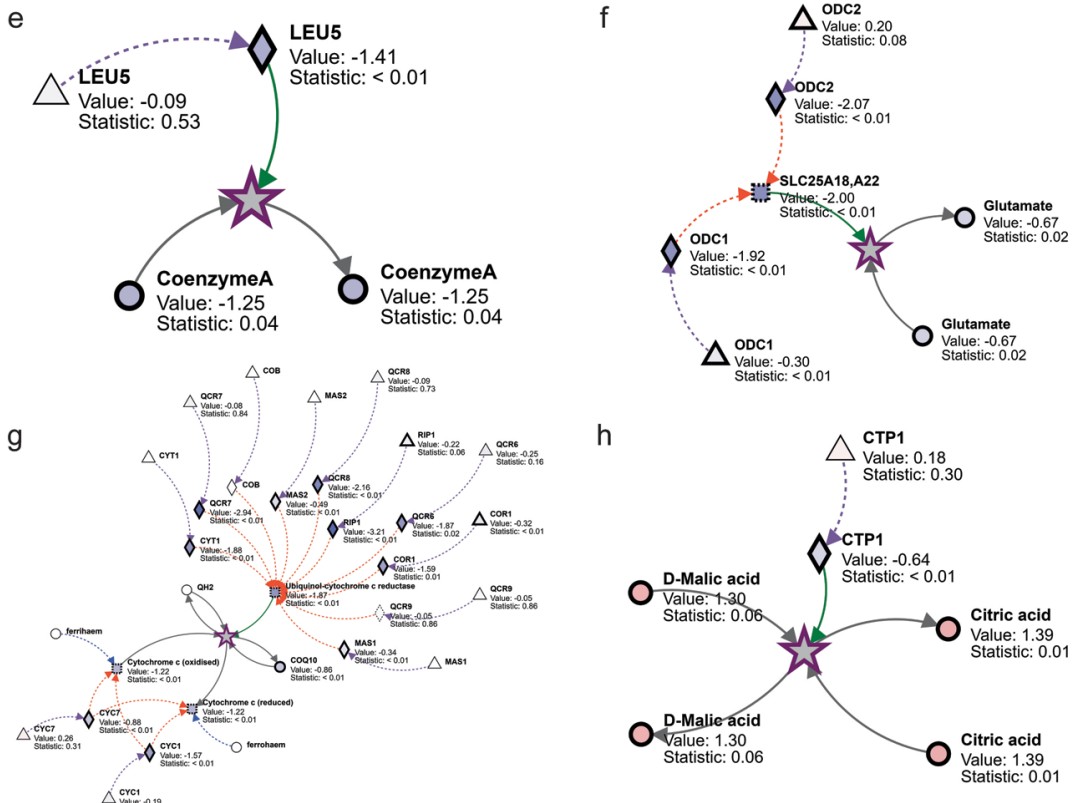

## Top TransReg Reaction Patterns

**Extended Data Fig. 7 | See next page for caption.**

**Extended Data Fig. 7 | Top-ranking** ModReg **and** TransReg **reaction patterns identified in steady-state proteomics and metabolomics data in the** *mct1* **Δ background.** Stamp view snapshots of four of the top-ranking ModReg and four of the top-ranking TransReg reaction patterns in the *mct1* Δ vs. wild-type comparison using steady-state (12-hr) proteomics and metabolomics data and sorted by associated statistical values. In cases where the substrates, products, and modifiers were identical, only the first reaction pattern is shown. Reaction patterns were sorted by difference in magnitude of the different relevant components. Only results where the input/output and modifier were both statistically significant are shown. 12-hr RNA-sequencing comparisons contained n=4 in each group, 12-hr proteomics comparisons contained n = 3 in each group, and 12-hr metabolomics comparisons contained n = 3 in each each comparison group. **a.** ModReg reaction #1: propionyl-CoA + carnitine = > propionylcarnitine + CoA SH, **b.** ModReg reaction #2: alpha-ketoadipate + CoA SH + NAD+ = > glutaryl-CoA + $CO_2$ + NADH + H+, **c.** ModReg reaction #3: FXN:NFS1:ISD11:ISCU assembles 2Fe-2S iron-sulfur cluster, **d.** ModReg reaction #4: Succinate < = > Fumarate (with FAD redox reaction on enzyme), **e.** TransReg reaction #1: SLC25A16 transports cytosolic CoA SH to mitochondrial matrix, **f.** TransReg reaction #2: SLC25A18,A22 cotransport Glu, H+ from cytosol to mitochondrial matrix, **g.** TransReg reaction #4: Electron transfer from ubiquinol to cytochrome c of complex III, and **h.** TransReg reaction #6: Transport of Citrate from Mitochondrial Matrix to cytosol. Source numerical data are available[64].

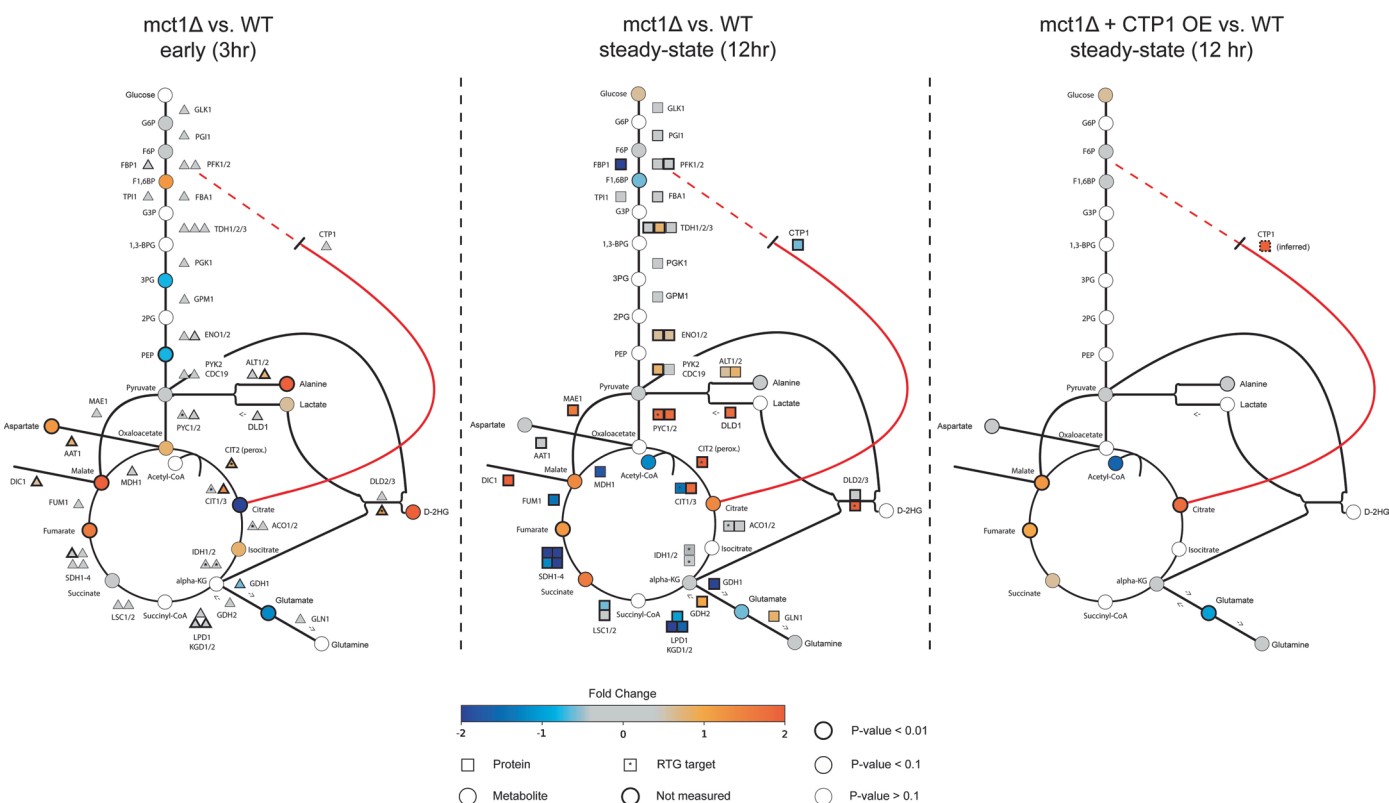

**Extended Data Fig. 8 | Graphical model of *mct1Δ* regulation.** Graphical overview of yeast glycolysis and the TCA cycle pathways and other related reactions overlaid with summary annotations based on RNA-sequencing, proteomics, and metabolomics measurements. Source numerical data are available[64].

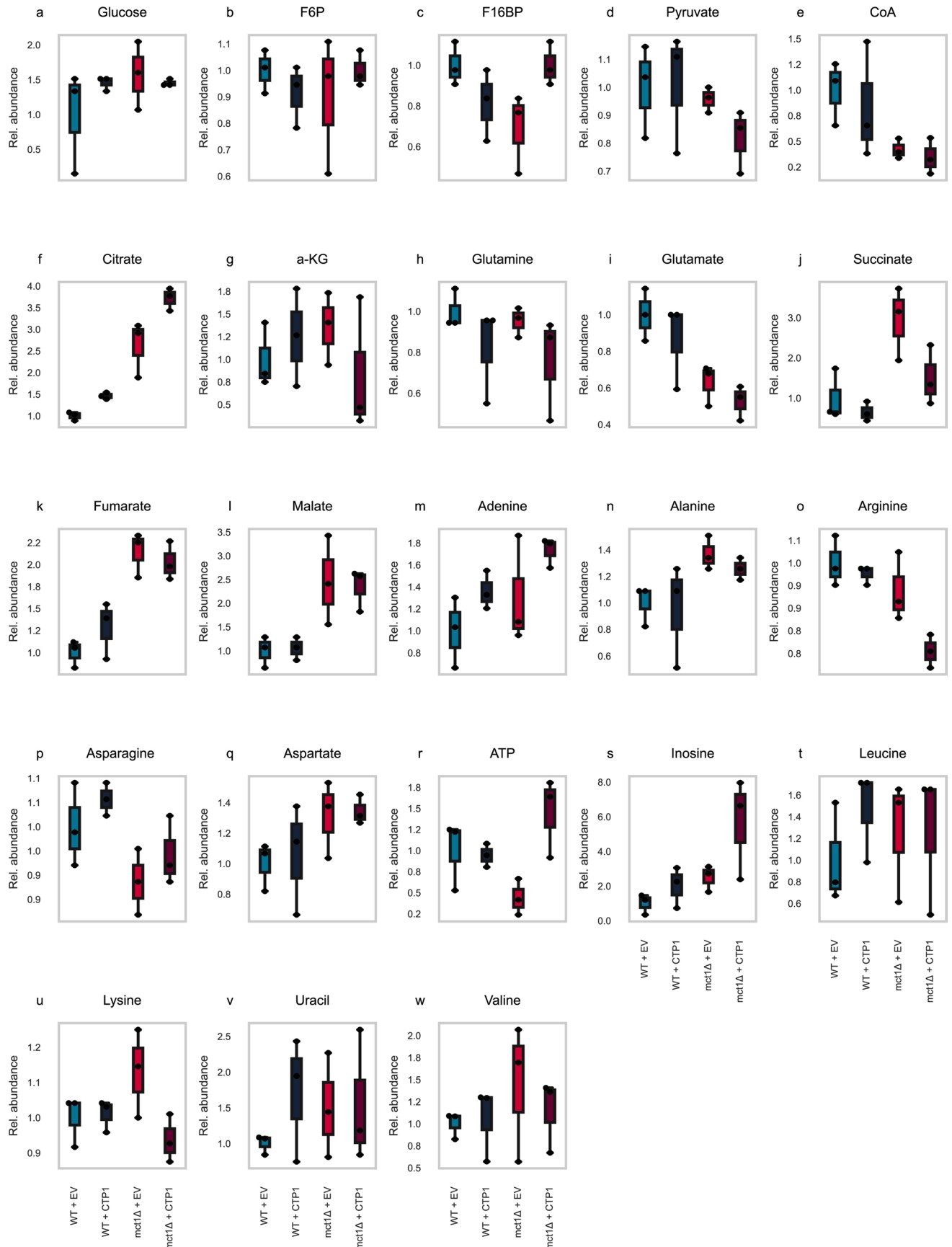

**Extended Data Fig. 9 | See next page for caption.**

**Extended Data Fig. 9 | Metabolite relative abundance changes during *CTP1* overexpression in *mct1* Δ or wild-type backgrounds at 12 hours growth in raffinose.** Boxplot overlaid with swarm plot for metabolites quantified by LC-MS in the *mct1* Δ or wild-type background with either an empty vector or vector overexpressing *CTP1* for **a**. glucose, **b**. fructose 6-phosphate (F6P), **c**. fructose 1,6-bisphosphate (F16BP), **d**. pyruvate, **e**. CoenzymeA species (CoA), **f**. citrate, **g**. *α*-ketoglutarate (a-KG), **h**. glutamine, **i**. glutamate, **j**. succinate, **k**. fumarate, **l**. malate, **m**. adenine, **n**. alanine, **o**. arginine, **p**. asparagine, **q**. aspartate, **r**. ATP, **s**. inosine, **t**. leucine, **u**. lysine, **v**. uracil, and **w**. valine. Cells were transferred from media with 2% glucose to media with 2% raffinose, allowed to grow for 12 hours, then harvested. All measurements were normalized using the average of the WT + EV samples for each metabolite. Each comparison group contained n = 3 samples. Center line represents data median, top and bottom lines represent 1.5x interquartile range. Individual data points are visualized as points. Source numerical data are available[64].

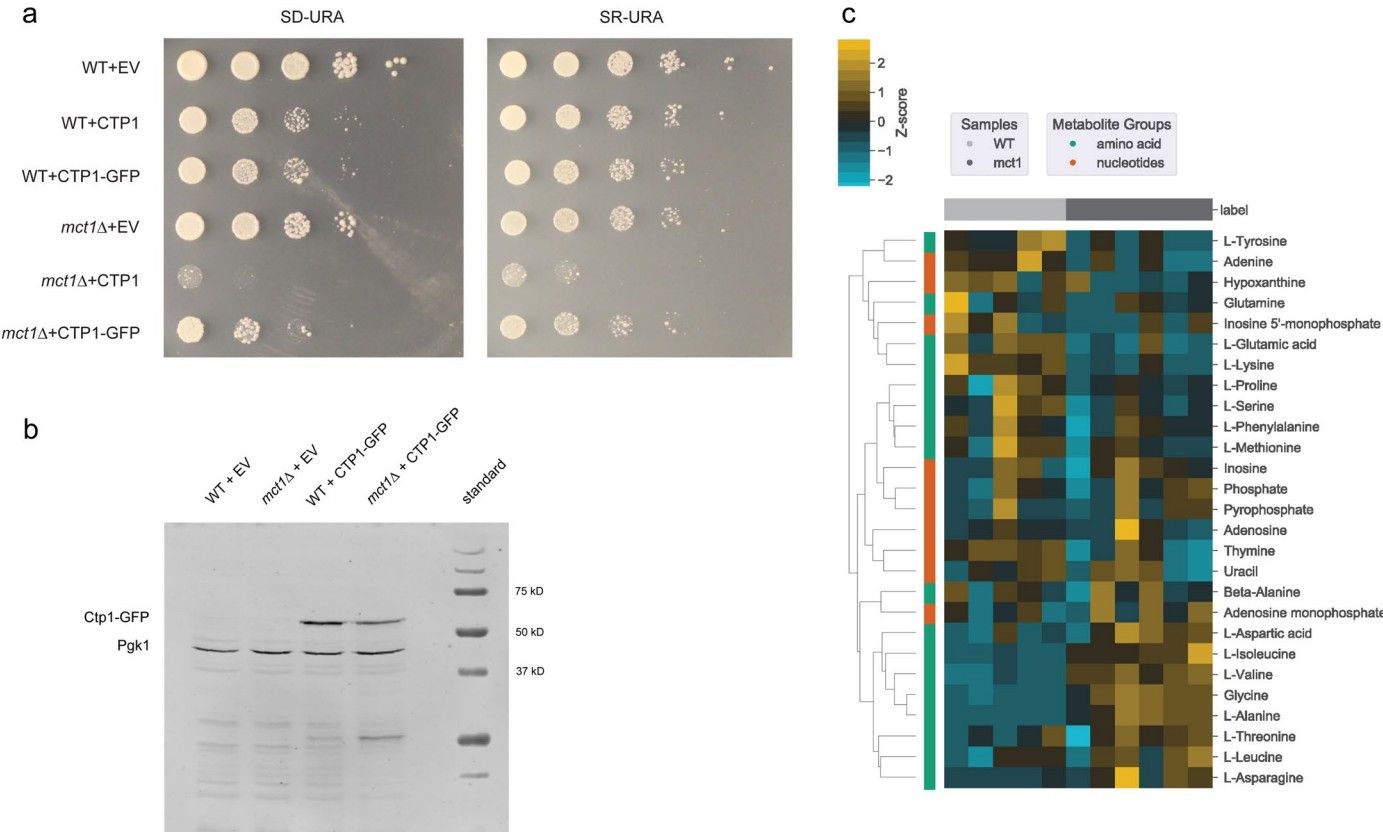

**Extended Data Fig. 10 | Supplemental data for the *mct1* Δ yeast model. a**. Spot dilutions of wild-type and *mct1* Δ yeast transformed with either empty vector (EV), *CTP1* overexpression (CTP1) vector, or *CTP1*-GFP fusion overexpression (CTP1-GFP) vector on synthetic media lacking uracil supplemented with either 2% glucose (left) or 2% raffinose (right). Cells were plated at mid-log phase (OD$_{600}$=0.3-0.6). Experiment was repeated 3 times. **b**. Western blots of wild-type and *mct1* Δ yeast transformed with either empty vector (EV), *CTP1* overexpression (CTP1) vector, or *CTP1*-GFP fusion overexpression (CTP1-GFP) vector. Experiment was repeated 1 time. **c**. Heatmap of amino acid and nucleotide metabolites for wild-type and *mct1* Δ mutant strain proteomics at 180 minutes post-raffinose carbon source shift. Metabolomics comparisons contained n = 6 in each each comparison group, except for the 3-hour wild-type group, which contained n = 5. Heatmap values were mean-centered at 0 (z-score). Hierarchical clustering was performed where indicated by the linkage lines using a simple agglomerative (bottom-up) hierarchical clustering method (or UPGMA (unweighted pair group method with arithmetic mean)). Source unprocessed blots, plate images, and numerical data are available[64].

# Reporting Summary

## Statistics

For all statistical analyses, confirm that the following items are present in the figure legend, table legend, main text, or Methods section.

| n/a | Confirmed | |
|---|---|---|
| ☐ | ☒ | The exact sample size (*n*) for each experimental group/condition, given as a discrete number and unit of measurement |
| ☐ | ☒ | A statement on whether measurements were taken from distinct samples or whether the same sample was measured repeatedly |
| ☐ | ☒ | The statistical test(s) used AND whether they are one- or two-sided *Only common tests should be described solely by name; describe more complex techniques in the Methods section.* |
| ☐ | ☒ | A description of all covariates tested |
| ☐ | ☒ | A description of any assumptions or corrections, such as tests of normality and adjustment for multiple comparisons |
| ☐ | ☒ | A full description of the statistical parameters including central tendency (e.g. means) or other basic estimates (e.g. regression coefficient) AND variation (e.g. standard deviation) or associated estimates of uncertainty (e.g. confidence intervals) |
| ☐ | ☒ | For null hypothesis testing, the test statistic (e.g. *F*, *t*, *r*) with confidence intervals, effect sizes, degrees of freedom and *P* value noted *Give P values as exact values whenever suitable.* |
| ☒ | ☐ | For Bayesian analysis, information on the choice of priors and Markov chain Monte Carlo settings |
| ☒ | ☐ | For hierarchical and complex designs, identification of the appropriate level for tests and full reporting of outcomes |
| ☐ | ☒ | Estimates of effect sizes (e.g. Cohen's *d*, Pearson's *r*), indicating how they were calculated |

*Our web collection on statistics for biologists contains articles on many of the points above.*

## Software and code

Policy information about availability of computer code

| Data collection | Metaboverse network files were prepared using Metaboverse v0.9.0 and v0.10.0, as specified in the manuscript. No other data collection or preparation software was used. |
|---|---|
| Data analysis | Metaboverse analysis was performed using Metaboverse v0.9.0 and v0.10.0, as specified in the manuscript. Additional source code generated for the analyses within this manuscript are available at https://github.com/Metaboverse/Metaboverse-manuscript/. Dependency version numbers are printed within code notebooks. List of software dependencies and versions (as available): Metaboverse (v0.9.0, v0.10.0) R (v4.0.3) survival (v3.2-11) survminer (v0.4.9) ggpubr (v0.4.0) XPRESSpipe (v0.6.0) DESeq2 (v1.22.1) dupRadar (v1.14.0) fastp (v0.20.0) star (v2.7.3a) bioconductor-rsamtools (v1.34.0) samtools (v1.10) |

```
bedtools (v2.29.2)
fastqc (v0.11.9)
htseq (v0.11.3)
matplotlib (v3.1.2, v3.4.2)
matplotlib-base (v3.1.1)
pandas (v1.0.2)
numpy (v1.17.4)
numpy-base (v1.17.4)
numpydoc (v0.9.2)
scipy (v1.4.1)
scikit-learn (v0.22.1)
multiqc (v1.8)
xpressplot (v0.2.4)
seaborn (v0.10.0, v0.11.0)
SpQN (v1.0.0)
Gene Ontology (GO) Resource (Release 2021-05-01)
PANTHER Overrepresentation Test (v16; Release 20210224)
El-MAVEN (v0.12.0)
MassHunter
Adobe Illustrator
Microsoft Excel
```

For manuscripts utilizing custom algorithms or software that are central to the research but not yet described in published literature, software must be made available to editors and reviewers. We strongly encourage code deposition in a community repository (e.g. GitHub). See the Nature Portfolio guidelines for submitting code & software for further information.

## Data

Policy information about availability of data

All manuscripts must include a data availability statement. This statement should provide the following information, where applicable:

- Accession codes, unique identifiers, or web links for publicly available datasets
- A description of any restrictions on data availability
- For clinical datasets or third party data, please ensure that the statement adheres to our policy

Gene expression counts for lung adenocarcinomas were obtained from the Human Protein Atlas project's TCGA FPKM gene expression data (https://www.proteinatlas.org/download/rna_cancer_sample.tsv.zip) and clinical patient data were obtained from TCGA (https://portal.gdc.cancer.gov/projects/TCGA-LUAD). Single cell data were obtained from the Human Lung Cell Atlas project v1.0 (https://zenodo.org/record/6337966#.YkzVrOjMIQ-). DepMap data 21Q4 Public was used.

mct1Δ and accompanying wild-type transcriptomics time-course data are deposited at the GEO repository under identifier GSE151606. mct1Δ and wild-type proteomics data are deposited at the ProteomeXchange repository under identifier PXD035000. Metabolomics data are deposited at the Metabolomics Workbench repository under project identifier PR000961, study identifier ST001401 and project identifier PR001422, study identifier ST002232. For gene co-expression analyses, all yeast samples available in refine.bio were accessed and downloaded on March 16, 2021.

## Human research participants

Policy information about studies involving human research participants and Sex and Gender in Research.

| Reporting on sex and gender | Not applicable |
| --- | --- |
| Population characteristics | Not applicable |
| Recruitment | Not applicable |
| Ethics oversight | Not applicable |

Note that full information on the approval of the study protocol must also be provided in the manuscript.

# Field-specific reporting

Please select the one below that is the best fit for your research. If you are not sure, read the appropriate sections before making your selection.

☒ Life sciences ☐ Behavioural & social sciences ☐ Ecological, evolutionary & environmental sciences

For a reference copy of the document with all sections, see nature.com/documents/nr-reporting-summary-flat.pdf

# Life sciences study design

All studies must disclose on these points even when the disclosure is negative.

| | |
|---|---|
| Sample size | For yeast experiments, samples were prepared with separate and fresh preparations with 3-6 biological replicates in each experimental or control group, as detailed in the Methods section and elsewhere as appropriate within the manuscript. In the case of the refine.bio yeast cohort, the entire wild-type sample cohort was used as specified in the manuscript text. For the public human lung adenocarinoma datasets, the Wikoff 2015 study contained 39 tumor tissue samples and 39 paired normal tissue samples; and TCGA data contained 487 gene expression samples total that were relevant to this study.<br><br>No statistical method was used to predetermine sample size. Sample sizes for high-throughput data generated for this study were chosen based on first-principles understanding of the number of samples needed to generate expected statistical distributions based on the data type. Statistical values were then adjusted for false positives following the convention for the respective data type. Other data were previously generated for other studies. |
| Data exclusions | For survival analysis, TCGA data were right censored and then removed if no days to death or censored days to death were available. Metabolomics samples that did not pass basic QC (n=1) were excluded from further analysis. No additional data were excluded. |
| Replication | All biological assays were repeated at least 3 times. All replication attempts were successful. Verification of plasmid construct expression by western blot was performed once as a simple validation that the construct was being over-expressed. |
| Randomization | Samples were randomized during sample preparation, but not during sample harvest for the yeast RNA sequencing and yeast metabolomics data, or were previously collected (human TCGA data, human metabolomics data, and yeast proteomics data), or otherwise not amenable to randomization (yeast growth spot tests, etc.). Yeast sample ordering and handling would have otherwise been randomized during sample processing. |
| Blinding | Data were either previously collected (human TCGA data, human metabolomics data, and yeast proteomics data), blinded during sample preparation but not sample harvest (yeast RNA sequencing, yeast metabolomics), or were otherwise not amenable to blinding (yeast growth spot tests, etc.). Yeast samples were additionally difficult to blind during growth and harvest as exact growth rates need to be measured throughout, and often correlate with genetic background. |

# Reporting for specific materials, systems and methods

We require information from authors about some types of materials, experimental systems and methods used in many studies. Here, indicate whether each material, system or method listed is relevant to your study. If you are not sure if a list item applies to your research, read the appropriate section before selecting a response.

## Materials & experimental systems

| n/a | Involved in the study |
|---|---|
| ☐ | ☒ Antibodies |
| ☒ | ☐ Eukaryotic cell lines |
| ☒ | ☐ Palaeontology and archaeology |
| ☒ | ☐ Animals and other organisms |
| ☒ | ☐ Clinical data |
| ☒ | ☐ Dual use research of concern |

## Methods

| n/a | Involved in the study |
|---|---|
| ☒ | ☐ ChIP-seq |
| ☒ | ☐ Flow cytometry |
| ☒ | ☐ MRI-based neuroimaging |

## Antibodies

| | |
|---|---|
| Antibodies used | α-GFP (Cell Signaling Technology #2956) and α-Pgk1 (Abcam #ab113687) |
| Validation | α-GFP (Cell Signaling Technology #2956): Rabbit. RRID: AB 1196615. 1/2000 dilution dilution used in this study. Applications: WB & IHC. Reactivity: Saccharomyces cerevisiae.  Datasheet available at https://www.cellsignal.com/datasheet.jsp?productId=2956&images=1<br>α-Pgk1 (Abcam #ab113687): Mouse Primary Monoclonal (22C5D8), Unconjugated, Unmodified. RRID: AB 10861977. 1/3000 dilution used in this study. Applications: WB & ICC-IF. Reactivity: Saccharomyces cerevisiae. No validations available, but cited many times in literature (https://www.citeab.com/antibodies/746015-ab113687-anti-pgk1-antibody-22c5d8). |

