## [Peer Review File · Nature Cell Biology]

Peer Review Information

Journal: Nature Cell Biology

Manuscript Title: Metaverse enables automated discovery and visualization of diverse metabolic regulatory patterns

Corresponding author name(s): Dr Jordan Berg

Editorial Notes:

Reviewer Comments & Decisions:

Decision Letter, initial version:

*Please delete the link to your author homepage if you wish to forward this email to co-authors.

Dear Jordan,

Thank you for submitting your manuscript, "Metaverse: Automated discovery and visualization of diverse metabolic regulatory patterns", to Nature Cell Biology and thank you for your patience with the peer review process. The manuscript has now been seen by 3 referees, who are experts in metabolomics, cancer, quantitative metabolism (Referee #1); network analyses in metabolism,

systems metabolism (Referee #2); and metabolomics, computer science, web tools (Referee #3). As you will see from their comments (attached below), they found this work of potential interest but have raised substantial concerns, which in our view would need to be addressed with considerable revisions before we can consider publication in Nature Cell Biology.

Nature Cell Biology editors discuss the referee reports in detail within the editorial team, including the chief editor, to identify key referee points that should be addressed with priority to strengthen the current conclusions. To guide the scope of the revisions, I have listed these points below. We are committed to providing a fair and constructive peer-review process, so please feel free to contact me if you would like to discuss any of the referee comments further. Our typical revision period is six months; please do let me know if you anticipate any delays or issues addressing the reviews, we are happy to discuss the revisions further.

In our view, for reconsideration at the journal, it would be essential to:

A- expand the test datasets to help readers evaluate the software, as suggested by Rev#1. Please also follow-up on the suggestions from this ref to validate the claims about CTP1.

B- address Revs#2 and #3's questions about the sensitivity of Metaboverse, the minimum dataset required and whether one can show that biological discoveries need the proper addressing of sparsity: -- Rev#2 first bulleted major point "The authors highlight the ability to collapse reactions in data sets with incomplete measurements and Metaboverse collapses up to three connection reactions. What happens if there are missing data points for more than 3 connected reactions? Consider providing sensitivity analysis? How much data is required to get reasonable results from Metaboverse (i.e. minimum number of metabolites, gene, and protein expression?)."

-- Rev#3 "Around line 138, it is described how the collapsing or reactions enables for a reduction in sparsity, is the level of collapse proposed (3 reactions) enough to cover a significant proportion of the metabolic network - given the standard analytical methods?"

-- Rev#3 "For the Yeast study, the authors have collected GC, LC HILIC, Proteomics, RNA Seq for the yeast. One of the major claims in this manuscript is that the proper addressing of sparsity will lead to less variable biological discoveries. The authors should put this to the test. What if only GC is used and LC HILIC were dropped, or vice versa, how does this change the ranking of the metabolites of interest? Similarly, what if we mask certain metabolites randomly from the raw data, similar results? Similar to above, what if you did LC RP? Would that change the results? Worth discussion, but I don't recommend doing actual wet lab experiments. Perhaps look for relevant public data."

C- expand the description of the Methods as suggested by Rev#3; please also address the reviewer's question about the benchmarking analyses, if it is possible to compare the presence of false positives using Metaboverse vs MNS/IPA

D- address all reviewers' remarks about statistical analyses, data presentation and discussion, the long-term plan for software maintenance, experimental questions, clarifications and textual changes. While we do not strictly require data deposition for metabolomics data, we ask that you please share as much as the data as possible as supplementary Table in excel form or deposit the data in a repository meeting the following criteria: <https://www.nature.com/sdata/policies/repositories>

E- Finally please pay close attention to our guidelines on statistical and methodological reporting (listed below) as failure to do so may delay the reconsideration of the revised manuscript. In particular please provide:

- a Supplementary Figure including unprocessed images of all gels/blots in the form of a multi-page pdf file. Please ensure that blots/gels are labeled and the sections presented in the figures are clearly indicated.
- a Supplementary Table including all numerical source data in Excel format, with data for different figures provided as different sheets within a single Excel file. The file should include source data giving rise to graphical representations and statistical descriptions in the paper and for all instances where the figures present representative experiments of multiple independent repeats, the source data of all repeats should be provided.

We would be happy to consider a revised manuscript that would satisfactorily address these points, unless a similar paper is published elsewhere or is accepted for publication in Nature Cell Biology in the meantime.

- ensure that it conforms to our format instructions and publication policies (see below and <https://www.nature.com/nature/for-authors>).
- provide a point-by-point rebuttal to the full referee reports verbatim, as provided at the end of this letter.
- provide the completed Reporting Summary (found here <https://www.nature.com/documents/nr-reporting-summary.pdf>). This is essential for reconsideration of the manuscript will be available to editors and referees in the event of peer review. For more information see <http://www.nature.com/authors/policies/availability.html> or contact me.

When submitting the revised version of your manuscript, please pay close attention to our [href="https://www.nature.com/nature-portfolio/editorial-policies/image-integrity">Digital Image Integrity Guidelines](https://www.nature.com/nature-portfolio/editorial-policies/image-integrity). and to the following points below:

Nature Cell Biology is committed to improving transparency in authorship. As part of our efforts in this direction, we are now requesting that all authors identified as 'corresponding author' on published papers create and link their Open Researcher and Contributor Identifier (ORCID) with their account on the Manuscript Tracking System (MTS), prior to acceptance. ORCID helps the scientific community achieve unambiguous attribution of all scholarly contributions. You can create and link your ORCID from the home page of the MTS by clicking on 'Modify my Springer Nature account'. For more information please visit www.springernature.com/orcid.

This journal strongly supports public availability of data. Please place the data used in your paper into a public data repository, or alternatively, present the data as Supplementary Information. If data can only be shared on request, please explain why in your Data Availability Statement, and also in the correspondence with your editor. Please note that for some data types, deposition in a public repository is mandatory - more information on our data deposition policies and available repositories appears below.

[Redacted]

We hope that you will find our referees' comments and editorial guidance helpful. Please do not hesitate to contact me if there is anything you would like to discuss. Thank you again for considering NCB for this work.

Best wishes,

Melina

Melina Casadio, PhD
Senior Editor, Nature Cell Biology
ORCID ID: <https://orcid.org/0000-0003-2389-2243>

Reviewers' Comments:

Reviewer #1:

Remarks to the Author:

In this manuscript, Berg, Zhou and colleagues design a software, Metaboverse, for automated discovery and visualization of metabolomics data. The focus is on finding patterns of potential reaction activity from metabolite profiling data when overlaid onto a chemical reaction network. Alternative approaches involving flux analysis require far more complicated experimentation and data analysis such as the use of heavy isotopes. The authors review existing analytical tools and compare

Metaboverse to them. Using published data in colon cancer, Metaboverse was able to identify potentially new changes in metabolism involving spermine synthase and glycerate kinase. The authors also studied the respiratory responses to mitochondrial fatty acid synthesis impairment by knocking out the MCT1 gene in yeast which previously has been shown to abolish mitochondrial fatty acid synthesis.

Overall, this is an important study potentially as it provides new a new tool to help biochemists identify possible non-intuitive metabolic activity from transcriptomic, proteomic, and metabolomic datasets. The software program is well-designed with detailed tutorials and documentation.

Comments:

- Line 211: It is hard to evaluate the software unless you process the most familiar multi-omics datasets with it. So, I encourage the author to provide more test datasets other than *saccharomyces cerevisiae*, preferably with nowadays more commonly used mammalian systems.
- Pages 13 to 17: The algorithm description on pages 13 to 17 may be better placed in the supporting information.
- Lines 215: The citrate level in the knockout group at 3 hours was decreasing. As the cell is adapting to a new culture medium in the first 3 hours, many pathways are affected, and it is risky to attribute the depletion of citrate to CTP1 only. Isotope tracing with U13-Glucose for the first 3 hours may provide a better understanding of the citrate depletion.
- Line 220: "We theorized that this response would maintain citrate pools within the mitochondrial matrix, where it is perhaps most physiologically important for these cells to be able to adapt to the loss of MCT1." Will "this response" alter the mitochondrial/cytosolic concentration gradient of citrate? How is the mitochondrial/cytosolic concentration gradient of citrate determined?

Minor:

- Supporting Data Fig. 8: Please clarify the incubation time (3 hours or 12 hours?) in the legend of supporting figure 8.
- Fig. 5d: Figure 5d was a little bit misleading as the time course x-axis is not evenly distributed. The first half of the x-axis stands for 30 mins while the second half of the x-axis stands for 150 mins.
- Supporting Data Fig. 12: What is the software or platform to generate supporting figure 12?
- After downloading the software (Metaboverse-v0.9.0) to my computer, I followed the introduction video to set up the analysis with the test data, however, it will stop at the "building network" page forever at "0%" and won't give me the analysis result. The problem was solved by responding to the following error: 'C:\XXX\XXX\Desktop\12-29-2021' is not recognized as an internal or external command operable program or batch file. It seems that I cannot name the folder as "12-29-2021".

Reviewer #2:

Remarks to the Author:

Metaboverse: Automated discovery and visualization of diverse metabolic regulatory patterns

The manuscript by Berg et al. presents a data visualization and analysis tool to explore metabolic reactions/relationships within a data set. Metaboverse allows the user to upload metabolomics, transcriptomic, and proteomic data that will be overlaid on a network database curated from existing metabolic reaction libraries (Reactome, BiGG, BioModels). The data is then processed to allow the user to dynamically explore metabolic regulatory patterns and generate hypotheses. Through the case

studies presented, the tool is useful to integrate multiple omics data set for hypothesis generation and design follow-up experiments.

The paper describes a useful approach for processing and metabolic interpretation of diverse biological datasets. This software may seem to fill a key need in the field – aiding in the interpretation of metabolic datasets. At times the language is a bit strong in giving credit to the software for elucidating mechanisms as an analytical tool. Rather, some aspects of the software suggest it is better framed as a hypothesis builder, since the omics data inputted can hardly serve as a focused experiment. This does not take away from the utility of identifying such correlations (and challenges overcome to obtain them), but such methodological approaches should be put in better context.

Major point(s):

- The authors highlight the ability to collapse reactions in data sets with incomplete measurements and Metaboverse collapses up to three connection reactions. What happens if there are missing data points for more than 3 connected reactions? Consider providing sensitivity analysis? How much data is required to get reasonable results from Metaboverse (i.e. minimum number of metabolites, gene, and protein expression?).
- Is this truly an “analytical tool” or more a pattern recognition or hypothesis generating software? The difference is subtle but analytical implies something directly measured.

Minor point(s):

- I think the manuscript would benefit from having an overall concept map or schematic of the case studies presented. The reaction diagrams are nice but a single concept map capturing the protein and metabolites involved would make the text much easier to follow.
- Provide list for gene/protein/metabolite symbols and abbreviations used throughout the manuscript (e.g. THF, OAA, GDA)
- Can Metaboverse work with just metabolomic/transcriptomic/proteomic data? Does the user always need to input combined omics data sets or are subsets enough?

Reviewer #3:

Remarks to the Author:

Berg et al. present a new computational tool called Metaboverse to prioritize and visualize multi-omics data in order to discover key metabolomic drivers in biology.

The problems addressed in this work are of significance for the community, specifically, 1. network based prioritization of metabolites/reactions of interest, 2. Addressing sparsity of measurement due to instrumentation limitations, 3. Integrated user interface with these functions to effectively explore and prioritize findings.

Reviewer Questions

It is noted in line 124, some prior work, I think it is worth citing or at examining and contextualizing the overlap of this work to this recently published work: Kellman, Benjamin P., Anne Richelle, Jeong-Yeh Yang, Digantkumar Chapla, Austin WT Chiang, Julia A. Najera, Chenguang Liang et al. "Elucidating human milk oligosaccharide biosynthetic genes through network-based multi-omics integration." Nature communications 13, no. 1 (2022): 1-15. While the applications differ, the idea of tackling the

sparsity of measurements in the metabolomics data has some commonality.

Around line 138, it is described how the collapsing or reactions enables for a reduction in sparsity, is the level of collapse proposed (3 reactions) enough to cover a significant proportion of the metabolic network - given the standard analytical methods?

In line 287, the comparison to MNS and IPA are definitely needed in this manuscript to demonstrate the ability to pull out relevant biological phenomena. In the presentation, it is noted to examine "verifiable or canonical signatures" which is a way to measure the sensitivity of the Metaboverse tool. However, it would be good to evaluate the presence of false positives, for example out of the top 10 pathways, how many are definitely not relevant. This reviewer recognizes the difficulty of this as there is so much undiscovered biology, but there might be some low hanging fruit that can help assess the specificity of the multiple tools.

Line 470, the algorithmic description in prose for under what circumstances to collapse is reasonable, but on first read, it is still quite confusing. It is somewhat shown in Figure 2, but it would likely be very helpful to describe each collapsing condition that is written in text also as a set of figures in the SI.

By collapsing reactions of the network to help address sparsity, I'm not sure if the claim that imputation isn't done at all is the best way to communicate this. I agree it might not be the more traditional imputation where missing values are explicitly substituted, but as the authors note, "a collapsed pseudo-reaction is created, summarizing all three reactions". This summarization and subsequent calculation of a score can implicitly infer some scores especially across different experiments where the set of missing metabolites could differ.

Throughout this manuscript, all statistical assessments use the t-test statistic and the properties of this are used to calculate FDR. Presumably this FDR and described as an FDR is shown to the user. There does not seem to be a discussion on the appropriateness of using a t-test on the data as metabolites may not be normally distributed, drastically skewing the accuracy of the p-value and FDR. Communication of this uncertainty is very important in slick GUI as users will take these at face value, rather than thinking of the list of metabolites/pathways as more or a prioritization list to follow up on. Ref on normality of metabolomics data: Vinaixa, M., Samino, S., Saez, I., Duran, J., Guinovart, J. J., & Yanes, O. (2012). A guideline to univariate statistical analysis for LC/MS-based untargeted metabolomics-derived data. *Metabolites*, 2(4), 775-795.

All metabolomics data (GC and LC) for the Yeast study are missing and should be deposited publicly in a repository such as Metabolomics Workbench, GNPS, or Metabolights.

Proteomics data for the yeast study is not available and should be deposited in a repository such as PRIDE or MassIVE.

The analytical methods for the proteomics data collection are not sufficient, for example, what instrument, what digestion, gradient, protein ID software, quantification software.

The software procedures for identifying metabolites in the LC HILIC data is under-described. What parameters were used in EI-Maven as feature finding/quant if highly variable depending on the processing. It was also not described if MS/MS data was collected but is inferred if NIST and Fiehn libraries were used.

The in-house metabolite library if it is MS/mS should be deposited publicly in a resource like MassBank or GNPS.

For the Yeast study, the authors have collected GC, LC HILIC, Proteomics, RNA Seq for the yeast. One of the major claims in this manuscript is that the proper addressing of sparsity will lead to less variable biological discoveries. The authors should put this to the test. What if only GC is used and LC HILIC were dropped, or vice versa, how does this change the ranking of the metabolites of interest?

Similarly, what if we mask certain metabolites randomly from the raw data, similar results?

Similar to above, what if you did LC RP? Would that change the results? Worth discussion, but I don't

recommend doing actual wet lab experiments. Perhaps look for relevant public data. Similar to above, what if GC derivatization instead covers a different set of metabolites? Discussion only, no need to run more wet lab experiments.

Minor Comments

This might be a small quibble from a style perspective, but as a fellow researcher that is in the game of developing bioinformatics methods and software meant for end-users with an emphasis on user-friendliness, I very much understand how important having users understand how to use the software is. However, I feel like reading the manuscript with UI hints, e.g. Page 4 line 106, providing these hints in-line with intuitive explanation of the methods is a bit counter-productive. I've always found putting these kinds of details as use-cases that people can walk through in the SI or as a separate youtube video served the purpose much better. Again, a style thing, but just something I've found to work well for my own work.

I'm personally curious about the maintenance plan for the software? Is it going to be supported by the first author or plan for community support? I've seen so much software die over the years as graduate students move on.

Overall, I think this work tackles three challenges the community is interested in and is put together in a nice package. While this author has not attempted to run the software, it at least superficially has taken the care to make it more accessible to a wider audience which is a good thing to see if it actually works on real biological problems. Overall, there are certain claims and clarifications in the manuscript that are necessary to be more complete and accessible for the community. I wish the authors the best of luck and looking forward to their resubmission.

TITLE – should be no more than 100 characters including spaces, without punctuation and avoiding

technical terms, abbreviations, and active verbs..

Methods should be written concisely, but should contain all elements necessary to allow interpretation and replication of the results. As a guideline, Methods sections typically do not exceed 3,000 words. The Methods should be divided into subsections listing reagents and techniques. When citing previous methods, accurate references should be provided and any alterations should be noted. Information

must be provided about: antibody dilutions, company names, catalogue numbers and clone numbers for monoclonal antibodies; sequences of RNAi and cDNA probes/primers or company names and catalogue numbers if reagents are commercial; cell line names, sources and information on cell line identity and authentication. Animal studies and experiments involving human subjects must be reported in detail, identifying the committees approving the protocols. For studies involving human subjects/samples, a statement must be included confirming that informed consent was obtained. Statistical analyses and information on the reproducibility of experimental results should be provided in a section titled "Statistics and Reproducibility".

All Nature Cell Biology manuscripts submitted on or after March 21 2016 must include a Data availability statement as a separate section after Methods but before references, under the heading "Data Availability". For Springer Nature policies on data availability see <http://www.nature.com/authors/policies/availability.html>; for more information on this particular policy see <http://www.nature.com/authors/policies/data/data-availability-statements-data-citations.pdf>. The Data availability statement should include:

- Accession codes for primary datasets (generated during the study under consideration and designated as "primary accessions") and secondary datasets (published datasets reanalysed during the study under consideration, designated as "referenced accessions"). For primary accessions data should be made public to coincide with publication of the manuscript. A list of data types for which submission to community-endorsed public repositories is mandated (including sequence, structure, microarray, deep sequencing data) can be found here <http://www.nature.com/authors/policies/availability.html#data>.
- Unique identifiers (accession codes, DOIs or other unique persistent identifier) and hyperlinks for datasets deposited in an approved repository, but for which data deposition is not mandated (see here for details <http://www.nature.com/sdata/data-policies/repositories>).
- At a minimum, please include a statement confirming that all relevant data are available from the authors, and/or are included with the manuscript (e.g. as source data or supplementary information), listing which data are included (e.g. by figure panels and data types) and mentioning any restrictions on availability.
- If a dataset has a Digital Object Identifier (DOI) as its unique identifier, we strongly encourage including this in the Reference list and citing the dataset in the Methods.

We recommend that you upload the step-by-step protocols used in this manuscript to the Protocol Exchange. More details can found at www.nature.com/protocolexchange/about.

All imaging data should be accompanied by scale bars, which should be defined in the legend. Cropped images of gels/blots are acceptable, but need to be accompanied by size markers, and to retain visible background signal within the linear range (i.e. should not be saturated). The boundaries of panels with low background have to be demarked with black lines. Splicing of panels should only be considered if unavoidable, and must be clearly marked on the figure, and noted in the legend with a statement on whether the samples were obtained and processed simultaneously. Quantitative comparisons between samples on different gels/blots are discouraged; if this is unavoidable, it should only be performed for samples derived from the same experiment with gels/blots were processed in parallel, which needs to be stated in the legend.

- For line art, graphs, charts and schematics we prefer Adobe Illustrator (.AI), Encapsulated PostScript (.EPS) or Portable Document Format (.PDF). Files should be saved or exported as such directly from the application in which they were made, to allow us to restyle them according to our journal house style.
- We accept PowerPoint (.PPT) files if they are fully editable. However, please refrain from adding PowerPoint graphical effects to objects, as this results in them outputting poor quality raster art. Text used for PowerPoint figures should be Helvetica (preferred) or Arial.
- We do not recommend using Adobe Photoshop for designing figures, but we can accept Photoshop generated (.PSD or .TIFF) files only if each element included in the figure (text, labels, pictures, graphs, arrows and scale bars) are on separate layers. All text should be editable in 'type layers' and line-art such as graphs and other simple schematics should be preserved and embedded within 'vector smart objects' - not flattened raster/bitmap graphics.
- Some programs can generate Postscript by 'printing to file' (found in the Print dialogue). If using an application not listed above, save the file in PostScript format or email our Art Editor, Allen Beattie for advice (a.beattie@nature.com).

The total number of Supplementary Figures (not including the “unprocessed scans” Supplementary Figure) should not exceed the number of main display items (figures and/or tables (see our Guide to Authors and March 2012 editorial <http://www.nature.com/ncb/authors/submit/index.html#suppinfo>; <http://www.nature.com/ncb/journal/v14/n3/index.html#ed>). No restrictions apply to Supplementary Tables or Videos, but we advise authors to be selective in including supplemental data.

GUIDELINES FOR EXPERIMENTAL AND STATISTICAL REPORTING

REPORTING REQUIREMENTS – We are trying to improve the quality of methods and statistics reporting in our papers. To that end, we are now asking authors to complete a reporting summary that collects information on experimental design and reagents. The Reporting Summary can be found here <https://www.nature.com/documents/nr-reporting-summary.pdf> If you would like to reference the guidance text as you complete the template, please access these flattened versions at <http://www.nature.com/authors/policies/availability.html>.

Author Rebuttal to Initial comments
--

Reviewers' Comments:

Reviewer #1:

Remarks to the Author:

In this manuscript, Berg, Zhou and colleagues design a software, Metaboverse, for automated discovery and visualization of metabolomics data. The focus is on finding patterns of potential reaction activity from metabolite profiling data when overlaid onto a chemical reaction network. Alternative approaches involving flux analysis require far more complicated experimentation and data analysis such as the use of heavy isotopes. The authors review existing analytical tools and compare Metaboverse to them. Using published data in colon cancer, Metaboverse was able to identify potentially new changes in metabolism involving spermine synthase and glycerate kinase. The authors also studied the respiratory responses to mitochondrial fatty acid synthesis impairment by knocking out the MCT1 gene in yeast which previously has been shown to abolish mitochondrial fatty acid synthesis.

Overall, this is an important study potentially as it provides new a new tool to help biochemists identify possible non-intuitive metabolic activity from transcriptomic, proteomic, and metabolomic datasets. The software program is well-designed with detailed tutorials and documentation.

We thank you for your kind comments.

Comments:

- Line 211: It is hard to evaluate the software unless you process the most familiar multi-omics datasets with it. So, I encourage the author to provide more test datasets other than *saccharomyces cerevisiae*, preferably with nowadays more commonly used mammalian systems.

We agree with the reviewer and have added a variety of additional datasets from other organisms and multi-omics combinations, as well as vignettes in the documentation, to assist users in learning to use Metaboverse. These datasets include a variety of multi-omics, single-omics, and/or timecourse datasets from human, mouse, zebrafish, and yeast. These example datasets will be distributed with each version of Metaboverse from v0.10.1 onward. The documentation describing these distributed datasets can be found at: <https://metaboverse.readthedocs.io/en/latest/content/vignettes.html>.

- Pages 13 to 17: The algorithm description on pages 13 to 17 may be better placed in the supporting information.

We appreciate this suggestion. We have moved the referenced information to Supplemental Text 1.

- Lines 215: The citrate level in the knockout group at 3 hours was decreasing. As the cell is adapting to a new culture medium in the first 3 hours, many pathways are affected, and it is risky to attribute the depletion of citrate to CTP1 only. Isotope tracing with U13-Glucose for the first 3 hours may provide a better understanding of the citrate depletion.

We worry that the phrasing of our hypothesis in the original draft of the manuscript was overly complicated and unclear. We want to clarify that Metaboverse focused our attention on the citrate export reaction facilitated by

CTP1, where other analytic tools have missed this feature. In looking at the behavior of citrate across the time course we measured, we noticed that citrate levels are initially depleted and then subsequently repleted. While these two features may indeed be related, the main takeaway we hope to leave with the reader is that *mct1Δ* cells are adapting in ways that support the maintenance of citrate abundance, and repression of CTP1 gene expression could be a component of that. We observe that *mct1Δ* cells appear to proliferate normally despite their respiratory defects, but overexpression of CTP1 disrupts this growth-enabling adaptation (Fig 5f). We have worked to distill the writing of this section to be more direct and to more clearly communicate the hypothesis we made using the dataset and Metaboverse.

We appreciate this suggestion from the reviewer and have performed $U\text{-}^{13}\text{C}$ glucose tracing on the wild-type and *mct1Δ* yeast strains at an earlier time point and at a later time point as requested. Since we wanted to avoid glucose repression, we spiked in $U\text{-}^{13}\text{C}$ glucose for 15 min and 30 min at 0.02% final concentration. We observed that, at both 1 hr and 8 hrs of growth, *mct1Δ* yeast appears to incorporate more ^{13}C label into citrate (M+2 isotopomer) from glucose compared to wild-type. This is seemingly at odds with the observation that *mtFAS*-deficient cells are also PDH-deficient (see <https://pubmed.ncbi.nlm.nih.gov/11356580/> and <https://pubmed.ncbi.nlm.nih.gov/8444795/>, which we now reference in the section “Metaboverse assists in identifying the downregulation of the mitochondrial citrate transporter during mitochondrial fatty acid synthesis impairment”). However, levels of citrate could be compensated through PYC (Fig 5i,k). We also measured tracing into succinate and fumarate and also noticed increased labeling, despite the associated respiratory defects. This leads us to hypothesize that along with the activation of anaplerotic pathways in *mct1Δ* yeast, a variety of other metabolic adaptations are employed that we do not yet understand, and, as we describe in our response to this reviewer’s next question, we cannot measure subcellular citrate concentrations in yeast. However, this vignette illustrates that Metaboverse was able to help the experimentalist design actionable hypotheses from their data. We posit that fully defining unexpected metabolic adaptations is outside of the scope of the manuscript.

U-13C6 glucose tracing in wild-type and *mct1Δ* yeast of citrate. Three biological replicates were measured for each condition, timepoint, and tracing experiment. Relative isotopic incorporation was measured for each timepoint by comparing to the average of that time point’s mean. Isotopic incorporation from glucose into citrate was measured by analyzing the M+2 isotopologue of citrate.

U-13C6 glucose tracing in wild-type and mct1Δ yeast of other TCA cycle intermediates. Three biological replicates were measured for each condition, timepoint, and tracing experiment. Relative isotopic incorporation was measured for each timepoint by comparing to the average of that time point's mean. Isotopic incorporation from glucose into citrate was measured by analyzing the M+2 isotopologue of citrate.

The raw data for the tracing experiment is being deposited to Metabolomics Workbench. Analysis of this data was performed by Graphpad Prism and using custom scripts available at: https://github.com/Metaboverse/Metaboverse-manuscript/tree/main/mct1_analysis. Retention times used in quantification are provided as Suppl. Table 3.

Summary of uploaded data sets

Please select an appropriate Datatrack ID from the table below to upload additional raw data files or select an appropriate mwTab Filename to edit metadata and results for already registered data.

DataTrack ID (upload raw data)	Study ID	Date Submitted	Data Type	mwTab FileName (edit study)	Archive Filename	User Comments	Data Review Status	Data Review Comments	Uploaded Files
3591	-	2022-11-22	Target edMS	j_berg_20221122_151218_mwt_ab.txt	mct1-tracing.zip	-	-	-	mct1-tracing.zip (2.1G)

• Line 220: “We theorized that this response would maintain citrate pools within the mitochondrial matrix, where it is perhaps most physiologically important for these cells to be able to adapt to the loss of MCT1.” Will “this

response” alter the mitochondrial/cytosolic concentration gradient of citrate? How is the mitochondrial/cytosolic concentration gradient of citrate determined?

*In order to quantify mitochondrial matrix versus cytosolic citrate pools in these backgrounds, we would need to perform MITO-tag isolation of mitochondria (<https://pubmed.ncbi.nlm.nih.gov/27565352/>) – in fact we have utilized this approach in the past (<https://pubmed.ncbi.nlm.nih.gov/33333007/>). However, due to the cell wall characteristic of yeast, this method is not tractable for our model organism as it takes much longer than ideal to digest the cell wall, which will lead to unreliable results. We do, however, postulate that something about the citrate axis of *mct1Δ* metabolism is changing. Many of these hypotheses are described as a part of Fig. 5 of the manuscript.*

Minor:

- Supporting Data Fig. 8: Please clarify the incubation time (3 hours or 12 hours?) in the legend of supporting figure 8.

We appreciate the reviewer pointing out this omission. These data were from 12 hours post-media change to nonfermentable media (2% glucose to 2% raffinose). The figure legend and figure title have been updated.

- Fig. 5d: Figure 5d was a little bit misleading as the time course x-axis is not evenly distributed. The first half of the x-axis stands for 30 mins while the second half of the x-axis stands for 150 mins.

We agree with the reviewer that the previous formatting of Fig. 5d (now 5e) was difficult to interpret. We have updated the figure to include an x-axis that is broken between time points of unequal difference.

- Supporting Data Fig. 12: What is the software or platform to generate supporting figure 12?

The pathway map from Fig. S12 was manually created using Adobe Illustrator. The hex codes for the fold change values were calculated in Python using Matplotlib. We have added these details to the Methods section under “Heatmaps, box plots, and other visualizations.” The code used to generate these hex values are archived at <https://doi.org/10.5281/zenodo.7396832> and will be distributed with the final publication as an additional supplemental file.

- After downloading the software (Metaboverse-v0.9.0) to my computer, I followed the introduction video to set up the analysis with the test data, however, it will stop at the “building network” page forever at “0%” and won’t give me the analysis result. The problem was solved by responding to the following error:
'C:\XXX\XXX\Desktop\12-29-2021' is not recognized as an internal or external command operable program or batch file. It seems that I cannot name the folder as “12-29-2021”.

We attempted to reproduce the error described above. Unfortunately, we were unable to replicate this issue. We documented our attempts to replicate this issue at <https://github.com/Metaboverse/Metaboverse/issues/118>.

Reviewer #2:

Remarks to the Author:

Metaboverse: Automated discovery and visualization of diverse metabolic regulatory patterns

The manuscript by Berg et al. presents a data visualization and analysis tool to explore metabolic reactions/relationships within a data set. Metaboverse allows the user to upload metabolomics, transcriptomic, and proteomic data that will be overlaid on a network database curated from existing metabolic reaction libraries (Reactome, BiGG, BioModels). The data is then processed to allow the user to dynamically explore metabolic regulatory patterns and generate hypotheses. Through the case studies presented, the tool is useful to integrate multiple omics data set for hypothesis generation and design follow-up experiments.

The paper describes a useful approach for processing and metabolic interpretation of diverse biological datasets. This software may seem to fill a key need in the field – aiding in the interpretation of metabolic datasets. At times the language is a bit strong in giving credit to the software for elucidating mechanisms as an analytical tool. Rather, some aspects of the software suggest it is better framed as a hypothesis builder, since the omics data inputted can hardly serve as a focused experiment. This does not take away from the utility of identifying such correlations (and challenges overcome to obtain them), but such methodological approaches should be put in better context.

We appreciate your constructive feedback regarding the framing of Metaboverse. We addressed this point throughout the revised manuscript. While by no means exhaustive, we list several examples here to try to exemplify modifications to the language we have made:

- *“These particular connections were missed in the original study, but were quickly and automatically highlighted to the user for further investigation with Metaboverse’s reaction pattern identification module. Thus, we are able to demonstrate Metaboverse’s capability to efficiently identify such patterns and enable facile exploration and hypothesis generation from a dataset.” (line 139).*
- *“Therefore, the insights guided by Metaboverse provided context beyond what was available from previous analyses and led us to predict a new connection between a metabolic reaction and cancer patient survival.” (line 164)*
- *“Metaboverse provides a much-needed visual tool for exploring metabolic networks and guiding hypothesis generation using user data.” (line 320)*
- *Text of Figure legend 5 changed to read: “Metaboverse assists in identifying compensatory mechanisms to mitochondrial dysfunction from multi-omics data in yeast”*

Major point(s):

- The authors highlight the ability to collapse reactions in data sets with incomplete measurements and Metaboverse collapses up to three connection reactions. What happens if there are missing data points for more than 3 connected reactions? Consider providing sensitivity analysis? How much data is required to get reasonable results from Metaboverse (i.e. minimum number of metabolites, gene, and protein expression?).

We sincerely appreciate the suggestion to include sensitivity analyses, and found that doing so provided interesting insights into the capabilities of the Metaboverse software. To test the sensitivity of Metaboverse, we randomly dropped out 0-90% of metabolites and 0-60% of protein measurements from the multi-omics yeast dataset and 0-90% of metabolites from the lung adenocarcinoma metabolomics dataset. We then curated Metaboverse networks for these combinations of dropouts and exported reaction pattern tables for the relevant reaction pattern type for each dataset.

The results of this experiment are contained in the new manuscript section, “Metaboverse enables intuitive hypothesis generation, even from sparse data”, within the new Fig. 7, and within the Methods titled “Sensitivity analysis”. In summary, we noticed that with each dropout, there was a proportional decrease in the number of reaction patterns identified. However, reaction collapsing buffered this loss of information. We also evaluated within each of these analyte dropouts whether we could still identify key signatures reported within the analyses of these datasets earlier in the manuscript. We found again that reaction collapsing helped buffer these effects. We want to emphasize that Metaboverse is a hypothesis generation tool, which we hope we have clarified better based on these review suggestions. Metaboverse can use whatever data is provided; however sparse, to identify reactions of interest and aid the experimentalist in generating actionable hypotheses. While we do not want to make specific recommendations on how much data is required as Metaboverse can use any amount of data provided, the included sensitivity analysis should act as a good benchmark for those with a similar question.

In regard to your second point, we felt in designing the reaction collapse tool that collapsing a maximum of 3 sequential reactions made the most sense. This was largely based on some loose precedent from other metabolism-related studies (see <https://pubmed.ncbi.nlm.nih.gov/31367041/>). This approach allows for some stability in the number of reactions that can be detected, despite sparse data (Fig. 7). But we want to be cautious with this method and avoid over-interpreting or over-imputing the data. In the lung adenocarcinoma dataset, we can increase the number of possible reaction patterns from 239 / 2219 (10.77%) to 2160 / 3853

(56.06%). In the *mct1Δ* dataset, we can increase the number of possible reaction patterns from 172 / 604 (28.48%) to 662 / 986 (67.14%) (see section “Metaboverse enables intuitive hypothesis generation, even from sparse data”).

• Is this truly an “analytical tool” or more a pattern recognition or hypothesis generating software? The difference is subtle but analytical implies something directly measured.

We have fixed the language throughout the text to emphasize Metaboverse’s purpose to analyze patterns using user data and the metabolic network to guide hypothesis generation, and have outlined several examples above where we have modified the language to emphasize this focus. Additionally, in the Discussion section, we have emphasized this as the goal of the software by including the following text:

- *“These particular connections were missed in the original study, but were quickly and automatically highlighted to the user for further investigation with Metaboverse’s reaction pattern identification module. Thus, we are able to demonstrate Metaboverse’s capability to efficiently identify such patterns and enable facile exploration and hypothesis generation from a dataset.” (line 139).*
- *“Therefore, the insights guided by Metaboverse provided context beyond what was available from previous analyses and led us to predict a new connection between a metabolic reaction and cancer patient survival.” (line 164)*
- *“Metaboverse provides a much-needed visual tool for exploring multi-omics data in the context of the metabolic network and guiding hypothesis generation.” (line 320)*
- *Text of Figure legend 5 changed to read: “Metaboverse assists in identifying compensatory mechanisms to mitochondrial dysfunction from multi-omics data in yeast”*
- *“We expect Metaboverse to become a foundational pattern recognition and hypothesis generation tool and augment the user’s experience when analyzing their own data or pre-existing datasets. Metaboverse is flexible in that it allows multiple data layers to be integrated into an metabolic network analysis within many model organisms. Users will be able to frame their data in the context of the metabolic reaction network, discover new and interesting patterns, and design experiments with a more holistic mindset to learn how metabolic reactions contribute to the broader biological mechanisms within their model.” (line 351)*

Minor point(s):

• I think the manuscript would benefit from having an overall concept map or schematic of the case studies presented. The reaction diagrams are nice but a single concept map capturing the protein and metabolites involved would make the text much easier to follow.

*We appreciate this suggestion to aid the reader in more easily conceptualizing the main points of each figure. We have included schematics of the experimental set-ups of the human lung adenocarcinoma metabolomics dataset (Fig. 3a) and the yeast *mct1Δ* multi-omics experiment (Fig. 5a) and feel these will help the readers better visualize the background of these analyses.*

• Provide list for gene/protein/metabolite symbols and abbreviations used throughout the manuscript (e.g. THF, OAA, GDA)

We appreciate this suggestion and have included abbreviation tables as Suppl. Table 2.1-4.

• Can Metaboverse work with just metabolomic/transcriptomic/proteomic data? Does the user always need to input combined omics data sets or are subsets enough?

We apologize for not sufficiently describing the capabilities of Metaboverse in the manuscript and documentation. The user can provide any combination of transcriptomics and/or proteomics and/or metabolomics, or any combination of data that maps to a transcript/protein/metabolite name or ID.

We have sought to better emphasize these details in the text (line 60), documentation (<https://metaboverse.readthedocs.io/en/latest/content/overview.html#data-inputs>), and software (“Variables and Data” page within Metaboverse where data are uploaded - see red highlight in image below).

We also want to mention that in response to Reviewer #1, we have added additional test datasets with each distribution of Metaboverse. These datasets include a variety of omics datasets and combinations of omics datasets. The addition of these datasets and the corresponding vignettes in the documentation (<https://metaboverse.readthedocs.io/en/latest/content/vignettes.html>) should assist users and readers in better assessing the scope of data with which Metaboverse is able to work and becoming familiar with the software.

Reviewer #3:

Remarks to the Author:

Berg et al. present a new computational tool called Metaboverse to prioritize and visualize multi-omics data in order to discover key metabolomic drivers in biology.

The problems addressed in this work are of significance for the community, specifically, 1. network based prioritization of metabolites/reactions of interest, 2. Addressing sparsity of measurement due to instrumentation limitations, 3. Integrated user interface with these functions to effectively explore and prioritize findings.

Reviewer Questions

It is noted in line 124, some prior work, I think it is worth citing or at examining and contextualizing the overlap of this work to this recently published work: Kellman, Benjamin P., Anne Richelle, Jeong-Yeh Yang, Digantkumar Chapla, Austin WT Chiang, Julia A. Najera, Chenguang Liang et al. "Elucidating human milk oligosaccharide biosynthetic genes through network-based multi-omics integration." Nature communications 13, no. 1 (2022): 1-15. While the applications differ, the idea of tackling the sparsity of measurements in the metabolomics data has some commonality.

We appreciate the reviewer bringing this approach to our attention. From the Introduction and the Methods sections of the referenced paper, the authors state:

“Starting from a scaffold of all possible reactions, we used constraint-based modeling to reduce the network to a set of relevant reactions and most plausible HMO structures when not known to form the basis for a mechanistic model.”

“Summary network extraction from the Reduced Network. The summary network relates a heuristic selection of the most important reactions in the HMO biosynthesis network as measured by proportion of inclusion in the commonly high-performing models and enrichment in the commonly high-performing models relative to the background. Paths drawn from observed HMOs to the root lactose were scored for their aggregate importance. The top 5% of paths leading to each observed HMO were retained to form the summary network (see Supplementary Methods 4.4.3).”

While we agree that this approach is interesting, we do not feel it is directly applicable to the question of handling sparsity and may confuse readers of what Metaboverse’s reaction collapse algorithm is trying to accomplish. In this work, the authors use constraint-based modeling to derive the most applicable reactions from some large search space of possible reactions when the exact reaction network is unknown. In this case, the authors are attempting to impute the structure of the network, whereas in Metaboverse we want to take the predefined metabolic network and simplify portions of the network where measurements are missing. If the reviewer and editor feel strongly that the referenced paper is still applicable, we are happy to include it, but it is our current opinion that the method is too divergent from what Metaboverse is trying to accomplish.

Around line 138, it is described how the collapsing or reactions enables for a reduction in sparsity, is the level of collapse proposed (3 reactions) enough to cover a significant proportion of the metabolic network - given the standard analytical methods?

In the case of the lung adenocarcinoma dataset, we go from 239 / 2219 (10.77%) metabolic reactions with measurements across them to 2160 / 3853 (56.06%) metabolic reactions or collapsed reaction representations. In the mct1Δ 12-hour metabolomics and proteomics dataset, we go from 172 / 604 (28.48%) metabolic reactions with measurements across them to 662 / 986 (67.14%) metabolic reactions or collapsed reaction representations. In both cases, we are able to drastically improve the number of metabolic reactions that we can then perform pattern searching across. While for both datasets this still amounts to incomplete coverage across the metabolic network, we still feel strongly that we do not want to collapse more than 3 reactions at a time so that the inputs and outputs of a collapsed reaction can still be feasibly related, and there is at least some precedent for such a number (<https://pubmed.ncbi.nlm.nih.gov/31367041/>). We appreciate this question from the reviewer and feel it helps emphasize some of the considerations going on behind the scenes with reaction collapsing in Metaboverse. We have included these metrics in the revised manuscript with the other analyses analyzing data sparsity (line 293).

In line 287, the comparison to MNS and IPA are definitely needed in this manuscript to demonstrate the ability to pull out relevant biological phenomena. In the presentation, it is noted to examine “verifiable or canonical signatures” which is a way to measure the sensitivity of the Metaboverse tool. However, it would be good to evaluate the presence of false positives, for example out of the top 10 pathways, how many are definitely not relevant. This reviewer recognizes the difficulty of this as there is so much undiscovered biology, but there might be some low hanging fruit that can help assess the specificity of the multiple tools.

We have given this question and suggestion considerable thought as we worked through other requested revisions these past several months, and as the reviewer alludes to, we do not feel comfortable evaluating false positives based on defining some measure of relevance. We simply do not know enough about what is or is not relevant to make a comfortable distinction. However, we do feel that the point raised is important, as the reviewer suggests, to expand on the evaluation of Metaboverse compared to other tools. We have therefore sought to expound upon the discussion within the “Metaboverse identifies meaningful and verifiable patterns that existing tools overlook” section and have added the section, “Metaboverse enables intuitive hypothesis generation, even from sparse data.” Some specific points that we addressed in these sections that we feel are relevant to this request are the following:

- *“Another challenge we experienced in using this software was the sheer number of genes per result the user needed to parse. For example, when examining the top 10 results from this dataset in MNS, the*

provided gene lists ranged from 1-447 genes per result, with a mean of 58 genes per result (<https://doi.org/10.5281/zenodo.7396832>). The scope of these lists can greatly hamper the user's ability to generate actionable hypotheses from their dataset" (line 257).

- *In the section, "Metaboverse enables intuitive hypothesis generation, even from sparse data," we evaluate the ability of Metaboverse to identify key reaction signatures that we feel are most relevant to each analyzed datasets based on the past literature and other experiments we have performed as part of this manuscript. We observed that, especially with reaction collapsing, we are able to preserve many of these key signatures, even with data dropout (Fig. 7b, d).*

Line 470, the algorithmic description in prose for under what circumstances to collapse is reasonable, but on first read, it is still quite confusing. It is somewhat shown in Figure 2, but it would likely be very helpful to describe each collapsing condition that is written in text also as a set of figures in the SI.

We have changed the referenced graphic in Fig. 2b to better demonstrate flow of reaction collapse by labeling each reaction considered under a reaction collapse event, and how these reactions are translated in the collapsed reaction representation. We have also added Fig S1 to provide a visual guide for Algorithm #4, which explains the considerations and conditions being evaluated with each reaction collapse event.

By collapsing reactions of the network to help address sparsity, I'm not sure if the claim that imputation isn't done at all is the best way to communicate this. I agree it might not be the more traditional imputation where missing values are explicitly substituted, but as the authors note, "a collapsed pseudo-reaction is created, summarizing all three reactions". This summarization and subsequent calculation of a score can implicitly infer some scores especially across different experiments where the set of missing metabolites could differ.

We appreciate this caution from the reviewer and have modified the referenced text, figures, and documentation to better clarify these important points.

- *"Of note, Metaboverse reaction collapsing does not utilize traditional imputation methods, which may introduce unnecessary bias into data interpretation where smaller sample sizes are available and there is no strong prior with which to predict missing values" (line 103).*
- *New or updated figures: Updated Fig. 2b and added Suppl. Fig. 1.*
- *Documentation:*
<https://metaboverse.readthedocs.io/en/latest/content/walkthrough.html#reaction-collapsing>

Throughout this manuscript, all statistical assessments use the t-test statistic and the properties of this are used to calculate FDR. Presumably this FDR and described as an FDR is shown to the user. There does not seem to be a discussion on the appropriateness of using a t-test on the data as metabolites may not be normally distributed, drastically skewing the accuracy of the p-value and FDR. Communication of this uncertainty is very important in slick GUI as users will take these at face value, rather than thinking of the list of metabolites/pathways as more or a prioritization list to follow up on. Ref on normality of metabolomics data: Vinaixa, M., Samino, S., Saez, I., Duran, J., Guinovart, J. J., & Yanes, O. (2012). A guideline to univariate statistical analysis for LC/MS-based untargeted metabolomics-derived data. *Metabolites*, 2(4), 775-795.

We agree with the complexity of choosing the appropriate statistical methodology with mass spectrometry data. We appreciate the emphasis on how the GUI might mislead some users of the software. We have improved the documentation for the software and the hints within the software to assist users in pausing and better evaluating any special circumstances their data may require.

- <https://metaboverse.readthedocs.io/en/latest/content/other-features.html#data-formatting-help>
- *Within the "Format dataset" on the "Data and Variables" page within Metaboverse, we have provided more guidance on data formatting.*

1) Input unformatted data table:

Statistical Values: (normal) ▾

Adjusted P-value (normal): Sample comparisons will use a 2-group ANOVA comparison to calculate a base p-value, which assumes data are normally distributed, followed by a Benjamini-Hochberg (BH) p-value correction for multiple hypothesis correction. BH correction is applied here as it is not a conservative as a Bonferroni correction procedure and is thus generally better suited for exploratory data analysis.

P-value (normal): Sample comparisons will use a 2-group ANOVA comparison to calculate a base p-value, with no addition p-value correction afterwards. Assumes data are normally distributed.

Confidence Intervals: Confidence intervals will be calculated for each experimental/control group. This method uses the `jStat.normalci()` method, assuming the input data array are normally distributed.

If you click this button after a sample comparison has already been performed, the change in p-value handling will only be applied to new comparisons. If you want to apply the selected procedure to other previous comparisons, you will need to start the data processing over. You can do this by clicking the Refresh icon on the top left of this page (🔄), closing this window and re-opening the data formatter, or you can refresh the page by clicking "View" -> "Reload" or CTRL + R (on Windows/Linux) or CMD + R (on MacOS).

We have also included a new feature in the data formatting tool that will show the user the distributions within their data along with further hints and suggestions on how to approach their data. This feature is available within Metaboverse v0.10.1 and onward. We include an example screenshot of this feature below, and further documentation can be found at:

<https://metaboverse.readthedocs.io/en/latest/content/other-features.html#data-formatting-help>

All metabolomics data (GC and LC) for the Yeast study are missing and should be deposited publicly in a repository such as Metabolomics Workbench, GNPS, or Metabolights.

The GC-MS data accompanying this manuscript were previously archived on Metabolomics Workbench (project identifier PR000961, study identifier ST001401; <https://www.metabolomicsworkbench.org/data/DRCCMetadata.php?Mode=Study&StudyID=ST001401>). We have now additionally archived the referenced LC-MS data on Metabolomics Workbench (project identifier PR001422, study identifier ST002232; <https://www.metabolomicsworkbench.org/data/DRCCMetadata.php?Mode=Study&StudyID=ST002232>). These identifiers are included in the “Data Availability” section of the manuscript.

Proteomics data for the yeast study is not available and should be deposited in a repository such as PRIDE or MassIVE.

We appreciate this suggestion and agree these data should have been deposited when they were first published in Van Vranken, et al., Mol. Cell, 2018 (<https://pubmed.ncbi.nlm.nih.gov/30118679/>). We have uploaded these data to ProteomeXchange and can be accessed with identifier PXD035000 (<http://proteomecentral.proteomexchange.org/cgi/GetDataset?ID=PXD035000>). This updated identifier is now included in the manuscript under the “Data availability” section.

The analytical methods for the proteomics data collection are not sufficient, for example, what instrument, what digestion, gradient, protein ID software, quantification software.

The referenced data were initially described in Van Vranken, et al., Mol. Cell, 2018 (<https://pubmed.ncbi.nlm.nih.gov/30118679/>). We have summarized these methods to provide more background and a summary of the details of the methods of this original experiment (please see the “Proteomics analysis” section of the Methods).

The software procedures for identifying metabolites in the LC HILIC data is under-described. What parameters were used in EI-Maven as feature finding/quant if highly variable depending on the processing?

We have corrected this section of the methods to read: “Data were collected, metabolites were identified, and their peak area was recorded using EI-MAVEN software [82–84]. A pre-entered compound list of m/z values and corresponding metabolites was utilized to enable EI-Maven EIC extraction of all samples. Manual visual examination of peaks selected by EI-Maven was performed and misannotated peaks were manually corrected and exported as an Excel spreadsheet (Microsoft, Redmond WA), as described in [82, 84].” (Methods, line 247)

We have also included our reference LC-MS HILIC retention time table as Suppl. Table 3.

It was also not described if MS/MS data was collected but is inferred if NIST and Fiehn libraries were used. The in-house metabolite library if it is MS/mS should be deposited publicly in a resource like MassBank or GNPS.

All GC-MS analysis was performed with an Agilent 5977b GC-MS MSD-HES and an Agilent 7693A automatic liquid sampler. Standard 70 eV electron impact ionization was performed. Metabolite identity was established using retention time from an in-house metabolite library developed using pure purchased standards, and by mass spectrum from the NIST and Fiehn libraries. By its very nature electron impact data from GC-MS is fragmented and thus we don't perform MS on it – except for very rare, specialized cases; which was not the case for this particular experiment.

The in-house library is not something that is generally deposited as a resource; electron impact data has been standardized and collected for 40 years then deposited into NIST, and there is minimal utility in adding more. We use our library for retention time matching and this will vary from GC to GC depending on the method and

thus is not really useful unless someone is using the same exact method. The in-house library is integrated into the Agilent quantification software used by the University of Utah Metabolomics Core, where this experiment was performed.

We are able, however, to provide the retention times table used for quantifying the LC-MS datasets, which were performed within the lab and not by the Metabolomics Core (Suppl. Table 3).

For the Yeast study, the authors have collected GC, LC HILIC, Proteomics, RNA Seq for the yeast. One of the major claims in this manuscript is that the proper addressing of sparsity will lead to less variable biological discoveries. The authors should put this to the test. What if only GC is used and LC HILIC were dropped, or vice versa, how does this change the ranking of the metabolites of interest? Similarly, what if we mask certain metabolites randomly from the raw data, similar results? Similar to above, what if you did LC RP? Would that change the results? Worth discussion, but I don't recommend doing actual wet lab experiments. Perhaps look for relevant public data. Similar to above, what if GC derivatization instead covers a different set of metabolites? Discussion only, no need to run more wet lab experiments.

We really appreciate this suggestion. In response to the Reviewer #2, we performed sensitivity analysis on the two datasets from the vignettes of this manuscript (the yeast multi-omics experiment and the lung adenocarcinoma metabolomics experiment). We believe these data can be used to provide further insights into how sparsity and different mass spectrometry methods should be considered and evaluated with these datasets (see section "Metaboverse enables intuitive hypothesis generation, even from sparse data").

One point we want to again emphasize is that Metaboverse is a hypothesis generation tool. It approaches data analysis by evaluating reactions and not entire pathways. Metaboverse is able to use whatever data is provided; however sparse, to identify reactions of interest and aid the experimentalist in generating actionable hypotheses. This is emphasized to us from the provided sensitivity analysis (Fig. 7) when we randomly dropped out metabolomics and/or proteomics analytes and performed various reaction pattern analyses on these data. We see that in many cases, increasing amounts of randomly masked analytes more or less linearly corresponds with the number of reaction patterns identified. But, many key reaction features appear to be stably identified across masking experiments (Fig. 7b,d). These are important considerations for the experimentalist to consider, and we have added a discussion to the "Metaboverse enables intuitive hypothesis generation, even from sparse data" section of the manuscript to highlight these important considerations highlighted by this analysis.

We also received a suggestion at a recent conference to include some uncertainty/bias metric to help users understand if a certain pathway is measured more thoroughly than others to help them consider any patterns that they may be missing due to certain levels of sparsity in their data. While beyond the scope of the current manuscript, this approach may further help experimentalists in planning and analyzing their experiments. This suggestion is being tracked at <https://github.com/Metaboverse/Metaboverse/issues/125>.

Minor Comments

This might be a small quibble from a style perspective, but as a fellow researcher that is in the game of developing bioinformatics methods and software meant for end-users with an emphasis on user-friendliness, I very much understand how important having users understand how to use the software is. However, I feel like reading the manuscript with UI hints, e.g. Page 4 line 106, providing these hints in-line with intuitive explanation of the methods is a bit counter-productive. I've always found putting these kinds of details as use-cases that people can walk through in the SI or as a separate youtube video served the purpose much better. Again, a style thing, but just something I've found to work well for my own work.

We really appreciate this advice, and have tried to take this to heart by providing additional documentation and test datasets. We have included more clear language in the documentation in the walkthrough of Metaboverse, emphasizing the user can follow along with one of the provided test datasets. These vignettes are included at <https://metaboverse.readthedocs.io/en/latest/content/walkthrough.html>. As suggested by another reviewer, we have included additional test datasets from a variety of organisms and data types that are included with each

release of Metaverse. Examples of using these test datasets are also included in the documentation at <https://metaverse.readthedocs.io/en/latest/content/vignettes.html>.

I'm personally curious about the maintenance plan for the software? Is it going to be supported by the first author or plan for community support? I've seen so much software die over the years as graduate students move on.

*This is an excellent point, and we wholeheartedly agree with this point and the challenges facing academic software development. The first author has recently started a position at Altos Labs where they will be continuing to develop the Metaverse platform. We are very excited about this opportunity as Altos Labs will give Metaverse access to dedicated software developers to continue to harden and expand the Metaverse code as an open-source platform. An integral portion of the first author's research at Altos Labs will be developing improved network approaches for multi-omics datasets. These improved capabilities will be integrated into the Metaverse platform on a rolling basis. Other exciting collaborations have also emerged from Metaverse, such as ongoing work with Dr. Flavio Alves Lara at the Instituto Oswaldo Cruz to develop a compatible metabolic map for *M. leprae* and perform analysis of their multi-omics data within the Metaverse ecosystem.*

We have also included a page in the documentation to provide other potential developers with details about back-end and front-end development. This documentation will be updated throughout the software's life at <https://metaverse.readthedocs.io/en/latest/content/backend-development.html>.

Overall, I think this work tackles three challenges the community is interested in and is put together in a nice package. While this author has not attempted to run the software, it at least superficially has taken the care to make it more accessible to a wider audience which is a good thing to see if it actually works on real biological problems. Overall, there are certain claims and clarifications in the manuscript that are necessary to be more complete and accessible for the community. I wish the authors the best of luck and looking forward to their resubmission.

We thank you for your enthusiasm and constructive criticism!

Decision Letter, first revision:

Our ref: NCB-TR48235A

13th January 2023

Dear Dr. Berg,

Thank you for submitting your revised manuscript "Metaboverse: Automated discovery and visualization of diverse metabolic regulatory patterns" (NCB-TR48235A). It has now been seen by the original referees and their comments are below. The reviewers find that the paper has improved in revision, and therefore we'll be happy in principle to publish it in Nature Cell Biology, pending minor revisions to comply with our editorial and formatting guidelines.

Please note that the current version of your manuscript is in a PDF format. Could you please email us a copy of the file in an editable format (Microsoft Word or LaTeX), as we can not proceed with PDFs at this stage? Many thanks for your attention to this point.

With the Word file in-hand, we will begin performing detailed checks on your paper and will send you a checklist detailing our editorial and formatting requirements within 1-2 weeks. Please do not upload the final materials and make any revisions until you receive this additional information from us.

Thank you again for your interest in Nature Cell Biology. Please do not hesitate to contact me if you have any questions.

Sincerely,

Melina

Melina Casadio, PhD
Senior Editor, Nature Cell Biology
ORCID ID: <https://orcid.org/0000-0003-2389-2243>

Reviewer #1 (Remarks to the Author):

My concerns have been satisfactorily addressed.

Reviewer #2 (Remarks to the Author):

The authors have addressed my concerns and improved the paper significantly with their sensitivity analysis section, which I found to be interesting and informative.

Reviewer #3 (Remarks to the Author):

We thank the authors for addressing this reviewer's concerns and questions thoroughly.

I wish the authors the best of luck as well congratulate the first author with their new position.

Decision Letter, Final Checks

Our ref: NCB-TR48235A

2nd February 2023

Dear Dr. Berg,

Thank you for your patience as we've prepared the guidelines for final submission of your Nature Cell Biology manuscript, "Metaverse: Automated discovery and visualization of diverse metabolic regulatory patterns" (NCB-TR48235A). Please carefully follow the step-by-step instructions provided in the attached file, and add a response in each row of the table to indicate the changes that you have made. Please also check and comment on any additional marked-up edits we have proposed within the text. Ensuring that each point is addressed will help to ensure that your revised manuscript can be swiftly handed over to our production team.

In addition, please provide a word document version of your manuscript at your earliest convenience, in order for our team to complete some outstanding manuscript checks.

In recognition of the time and expertise our reviewers provide to Nature Cell Biology's editorial process, we would like to formally acknowledge their contribution to the external peer review of your manuscript entitled "Metaverse: Automated discovery and visualization of diverse metabolic regulatory patterns". For those reviewers who give their assent, we will be publishing their names alongside the published article.

Nature Cell Biology offers a Transparent Peer Review option for new original research manuscripts submitted after December 1st, 2019. As part of this initiative, we encourage our authors to support increased transparency into the peer review process by agreeing to have the reviewer comments, author rebuttal letters, and editorial decision letters published as a Supplementary item. When you

submit your final files please clearly state in your cover letter whether or not you would like to participate in this initiative. Please note that failure to state your preference will result in delays in accepting your manuscript for publication.

Cover suggestions

As you prepare your final files we encourage you to consider whether you have any images or illustrations that may be appropriate for use on the cover of Nature Cell Biology.

Nature Cell Biology has now transitioned to a unified Rights Collection system which will allow our Author Services team to quickly and easily collect the rights and permissions required to publish your work. Approximately 10 days after your paper is formally accepted, you will receive an email in providing you with a link to complete the grant of rights. If your paper is eligible for Open Access, our Author Services team will also be in touch regarding any additional information that may be required to arrange payment for your article.

Please note that *Nature Cell Biology* is a Transformative Journal (TJ). Authors may publish their research with us through the traditional subscription access route or make their paper immediately open access through payment of an article-processing charge (APC). Authors will not be required to make a final decision about access to their article until it has been accepted. Find out more about Transformative Journals

Authors may need to take specific actions to achieve compliance with funder and institutional open access mandates. If your research is supported by a funder that requires immediate open access (e.g. according to Plan S principles) then you should select the gold OA route, and we will direct you to the compliant route where possible. For authors selecting the subscription publication route, the journal's standard licensing terms will need to be accepted, including self-archiving policies. Those licensing terms will supersede any other terms that the author or any third party may assert apply to any version of the manuscript.

For information regarding our different publishing models please see our Transformative

Journals page. If you have any questions about costs, Open Access requirements, or our legal forms, please contact ASJournals@springernature.com.

Please use the following link for uploading these materials:
[Redacted]

Best regards,

Kendra Donahue
Staff
Nature Cell Biology

On behalf of

Melina Casadio, PhD
Senior Editor, Nature Cell Biology
ORCID ID: <https://orcid.org/0000-0003-2389-2243>

Reviewer #1:
Remarks to the Author:
My concerns have been satisfactorily addressed.

Reviewer #2:
Remarks to the Author:
The authors have addressed my concerns and improved the paper significantly with their sensitivity analysis section, which I found to be interesting and informative.

Reviewer #3:
Remarks to the Author:
We thank the authors for addressing this reviewer's concerns and questions thoroughly.

I wish the authors the best of luck as well congratulate the first author with their new position.

Final Decision Letter:

Dear Dr Berg,

I am pleased to inform you that your manuscript, "Metaverse enables automated discovery and visualization of diverse metabolic regulatory patterns", has now been accepted for publication in Nature Cell Biology.

Please note that *Nature Cell Biology* is a Transformative Journal (TJ). Authors may publish their research with us through the traditional subscription access route or make their paper immediately open access through payment of an article-processing charge (APC). Authors will not be required to make a final decision about access to their article until it has been accepted. Find out more about Transformative Journals

Authors may need to take specific actions to achieve compliance with funder and institutional open access mandates. If your research is supported by a funder that requires immediate open access (e.g. according to Plan S principles) then you should select the gold OA route, and we will direct you to the compliant route where possible. For authors selecting the subscription

publication route, the journal's standard licensing terms will need to be accepted, including self-archiving policies. Those licensing terms will supersede any other terms that the author or any third party may assert apply to any version of the manuscript.

If you have not already done so, we strongly recommend that you upload the step-by-step protocols used in this manuscript to the Protocol Exchange (www.nature.com/protocolexchange), an open online resource established by Nature Protocols that allows researchers to share their detailed experimental know-how. All uploaded protocols are made freely available, assigned DOIs for ease of citation and are fully searchable through nature.com. Protocols and Nature Portfolio journal papers in which they are used can be linked to one another, and this link is clearly and prominently visible in the online versions of both papers. Authors who performed the specific experiments can act as primary authors for the Protocol as they will be best placed to share the methodology details, but the Corresponding Author of the present research paper should be included as one of the authors. By uploading your Protocols to Protocol Exchange, you are enabling researchers to more readily reproduce or adapt the methodology you use, as well as increasing the visibility of your protocols and papers. You can also establish a dedicated page to collect your lab Protocols. Further information can be found at www.nature.com/protocolexchange/about

With kind regards,

Melina

Melina Casadio, PhD
Senior Editor, Nature Cell Biology
ORCID ID: <https://orcid.org/0000-0003-2389-2243>